# Translation affects mRNA stability in a codon-dependent manner in human cells

Qiushuang Wu, Santiago Gerardo Medina, Gopal Kushawah, Michelle Lynn DeVore, Luciana A Castellano, Jacqelyn M Hand, Matthew Wright, Ariel Alejandro Bazzini*

Stowers Institute for Medical Research, Kansas City, United States

**Abstract** mRNA translation decodes nucleotide into amino acid sequences. However, translation has also been shown to affect mRNA stability depending on codon composition in model organisms, although universality of this mechanism remains unclear. Here, using three independent approaches to measure exogenous and endogenous mRNA decay, we define which codons are associated with stable or unstable mRNAs in human cells. We demonstrate that the regulatory information affecting mRNA stability is encoded in codons and not in nucleotides. Stabilizing codons tend to be associated with higher tRNA levels and higher charged/total tRNA ratios. While mRNAs enriched in destabilizing codons tend to possess shorter poly(A)-tails, the poly(A)-tail is not required for the codon-mediated mRNA stability. This mechanism depends on translation; however, the number of ribosome loads into a mRNA modulates the codon-mediated effects on gene expression. This work provides definitive evidence that translation strongly affects mRNA stability in a codon-dependent manner in human cells.
DOI: https://doi.org/10.7554/eLife.45396.001

## Introduction

Messenger RNA (mRNA) degradation and mRNA translation represent two fundamental steps in the regulation of gene expression. Stability of the mRNA affects mRNA levels (*Herzog et al., 2017*) which in turn, impact protein production (*Ingolia, 2016*). Alterations in mRNA degradation leads to developmental defects (*Giraldez et al., 2006*) and human disease (*Goodarzi et al., 2016*). Likewise, aberrant mRNA translation has been implicated in protein misfolding and neurodegenerative disease (*Kapur et al., 2017*; *Pechmann and Frydman, 2013*), viral infection (*Walsh et al., 2013*) and developmental defects (*Gonskikh and Polacek, 2017*; *Kong and Lasko, 2012*). Recent studies have shown that translation impacts mRNA stability in a codon-dependent manner in yeast (*Harigaya and Parker, 2016*; *Presnyak et al., 2015*; *Radhakrishnan et al., 2016*), E. coli (*Boël et al., 2016*), zebrafish (*Bazzini et al., 2016*; *Mishima and Tomari, 2016*), Xenopus (*Bazzini et al., 2016*), Trypanosoma brucei (*de Freitas Nascimento et al., 2018*; *Jeacock et al., 2018*) and Drosophila melanogaster (*Burow et al., 2018*). This mechanism, termed codon optimality, refers to the ability of a given codon to affect mRNA stability in a translation-dependent manner (*Presnyak et al., 2015*). mRNAs enriched in 'optimal' codons tend to be more stable, display greater abundance, higher translation efficiency and longer poly(A)-tails. Conversely, mRNAs enriched in 'non-optimal' codons tend to be unstable, display lower homeostatic RNA levels, poor translation efficiency and shorter poly(A)-tails (*Bazzini et al., 2016*; *Mishima and Tomari, 2016*; *Radhakrishnan et al., 2016*; *Webster et al., 2018*).

Most efforts aimed at identifying *cis*-regulatory elements affecting mRNA stability have focused on sequences within 3' untranslated regions (UTRs). MicroRNAs and RNA binding proteins repress translation and/or destabilize target mRNAs through recognition of regulatory elements located primarily in the 3'UTR (*Bazzini et al., 2012*; *Despic and Neugebauer, 2018*; *Giraldez et al., 2006*;

*For correspondence:
arb@stowers.org

Competing interests: The authors declare that no competing interests exist.

**eLife digest** Proteins are made by joining together building blocks called amino acids into strings. The proteins are 'translated' from genetic sequences called mRNA molecules. These sequences can be thought of as series of 'letters', which are read in groups of three known as codons. Molecules called tRNAs recognize the codons and add the matching amino acids to the end of the protein. Each tRNA can recognize one or several codons, and the levels of different tRNAs inside the cell vary.

There are 61 codons that code for amino acids, but only 20 amino acids. This means that some codons produce the same amino acid. Despite this, there is evidence to suggest that not all of the codons that produce the same amino acid are exactly equivalent. In bacteria, yeast and zebrafish, some codons seem to make the mRNA molecule more stable, and others make it less stable. This might help the cell to control how many proteins it makes. It was not clear whether the same is true for humans.

To find out, Wu et al. used three separate methods to examine mRNA stability in four types of human cell. Overall, the results revealed that some codons help to stabilize the mRNA, while others make the mRNA molecule break down faster. The effect seems to depend on the supply of tRNAs that have a charged amino acid; mRNA molecules were more likely to self-destruct in cells that contained codons with low levels of the tRNA molecules.

Wu et al. also found that conditions in the cell can alter how strongly the codons affect mRNA stability. For example, a cell that has been infected by a virus reduces translation. Under these conditions, the identity of the codons in the mRNA has less effect on the stability of the mRNA molecule.

Changes to protein production happen in many diseases. Understanding what controls these changes could help to reveal more about our fundamental biology, and what happens when it goes wrong.

DOI: https://doi.org/10.7554/eLife.45396.002

*Meyer et al., 2012*; *Ray et al., 2013*; *Tadros et al., 2007*). However, on average, the coding sequence is approximately twice as long as the 3'UTR and is recognized by the most abundant RNA-binding complex in the cell: the ribosome (*Doudna and Rath, 2002*). While translating the information from the mRNA into protein is crucial, translation is also involved in quality control mechanisms of mRNAs (*Shoemaker and Green, 2012*). The mechanism of codon optimality represents a novel regulatory function of the ribosome upon properly processed mRNAs (*Richter and Coller, 2015*; *Schikora-Tamarit and Carey, 2018*; *Wu and Bazzini, 2018*).

Translation elongation rates underlie codon-mediated mRNA stability in yeast (*Hanson et al., 2018*; *Presnyak et al., 2015*). While less clear in higher organisms, optimal and non-optimal codons tend to be decoded by tRNAs that are highly or poorly expressed, respectively (*Bazzini et al., 2016*). This suggests that the supply and/or demand for specific tRNAs affects translation elongation, which in turn, affects mRNA stability (*Despic and Neugebauer, 2018*; *Richter and Coller, 2015*). The tRNA repertoire can fluctuate in different cell types (*Gingold et al., 2014*; *Goodarzi et al., 2016*) and/or following stress (*Torrent et al., 2018*). For example, proliferating cells undergoing differentiation contain tRNA profiles that correlate with codon usage of the specific transcriptome of each cell type (*Gingold et al., 2014*). Altering tRNA availability can lead to neurodegenerative diseases (*Ishimura et al., 2014*) and upregulation of specific tRNAs drives metastasis by enhancing stability of transcripts enriched in their cognate codons (*Goodarzi et al., 2016*).

To investigate whether translation affects mRNA stability in a codon-dependent manner in humans, we have measured the ability of each codon to affect mRNA stability. We have conducted analyses in four different human cell lines, using three independent methods to measure mRNA stability. We demonstrate that the regulatory information affecting mRNA stability is encoded specifically within codon identity, rather than other nucleotide sequence information. Destabilizing codons tend to have lower respective tRNA levels, as well as lower ratios of charged-tRNA (with amino acid) compared to stabilizing codons. Genes enriched in stabilizing codons also tend to possess longer poly(A)-tails than genes enriched in destabilizing codons. However, the poly(A)-tail is not essential

for affecting mRNA stability in a codon-dependent manner. We demonstrate that codon-mediated effect on gene expression can also be modulated by tuning the translational level (number of ribosomes on mRNA) through either global change in translation efficiency (e.g. viral infection) or specific UTRs sequences. Our studies reveal that in human cells, the ribosome interprets two codes within the mRNA: the genetic code, which specifies the amino acid sequence, and a 'codon-optimality-code', which shapes mRNA stability.

## Results

### Human mRNA decay correlates with codon composition in human cell lines

To determine whether codon composition affects mRNA stability in human cells, we treated 293T, HeLa and RPE cells with Actinomycin D to block transcription (*Figure 1—figure supplement 1A*) and measured decay of existing mRNAs by performing time-course mRNA-seq (*Figure 1A*). We calculated the codon stability coefficient (CSC) as the Pearson correlation coefficient between mRNA stability and codon occurrence. Based on this approach, we and others have previously determined the properties of each of 61 codons in mRNA stability in yeast (*Presnyak et al., 2015*), zebrafish, and *Xenopus* embryos (*Bazzini et al., 2016*) (*Figure 1A*). Codons displaying a positive correlation were referred to as 'optimal' codons because mRNAs enriched in those codons were more stable. Conversely, codons displaying a negative correlation were referred to as 'non-optimal' codons because mRNAs enriched in those codons were less stable (*Figure 1A, B*). These codon regulatory properties (optimal, non-optimal) correlate well across the three analyzed human cell types and zebrafish (*Figure 1C* and *Figure 1—figure supplement 1B*), suggesting that the regulatory activity of some codons is shared between humans and zebrafish. The CSC scores do not present strong correlation with codon usage (*Figure 1—figure supplement 1C*). These results also imply that codon composition has effects on mRNA stability in humans that are analogous to those found in other species.

mRNA stability can be influenced by *cis*-regulatory sequences within 5' and 3' UTRs of mRNAs which are targeted by microRNAs, RNA-binding proteins and RNA modification (m6A) (*Despic and Neugebauer, 2018*; *Meyer et al., 2012*). To separate the effects on stability related to codon composition from those due to *cis*-elements in the UTRs, we have previously developed a method to measure the regulatory properties of the coding sequence in a UTR-independent manner (*Bazzini et al., 2016*). We compared the decay of millions of exogenous mRNAs in zebrafish and *Xenopus* embryos which possess similar 5' and 3'UTRs but differ in codon composition (*Bazzini et al., 2016*). To adapt this strategy to human cells, we developed a method using a vector-based library, termed ORFome (*Yang et al., 2011*), containing ~17,000 different human coding sequences fused to common promoter, 5' and 3' UTRs (*Figure 1D*). ORFome library infection into 293T and K562 cells was followed by transcriptional inhibition (Actinomycin D treatment) and ORFome mRNA levels were monitored over time using small barcodes in the 3'UTR (*Figure 1D*). The distribution of mRNA level in the more than ten thousand ORFome genes is narrower than the endogenous mRNAs, likely because all ORFome mRNAs share the same strong promoter, 5' and 3' UTRs (*Figure 1E*). Nonetheless, the CSC scores derived from ORFome mRNAs correlated with respective endogenous mRNAs, further supporting that codon composition affects mRNA stability (*Figure 1B,C* and *Figure 1—figure supplement 1B*). To further explore that stability of the ORFome-derived mRNAs are indeed independent of UTR-mediated regulation, we selected mRNAs predicted to be targets of methylation (m6A) and compared that to a control group (*Meyer et al., 2012*). As expected, the endogenous m6A target mRNAs are more unstable than a control group (*Meyer et al., 2012*; *Yue et al., 2015*). However, there was significantly less difference between the same mRNAs when derived from ORFome, most likely due to the absence of methylation-sensitive 3'UTR regulatory elements (*Figure 1F*). Together, these observations suggest that this approach enables the dissection and measurement of regulatory activities embedded within coding regions and/or UTRs.

The above analyses depend on blocking transcription by Actinomycin D treatment, which may have unintended consequences. To circumvent this problem and to measure mRNA decay without blocking transcription, we employed SLAM-seq method (*Herzog et al., 2017*). This method

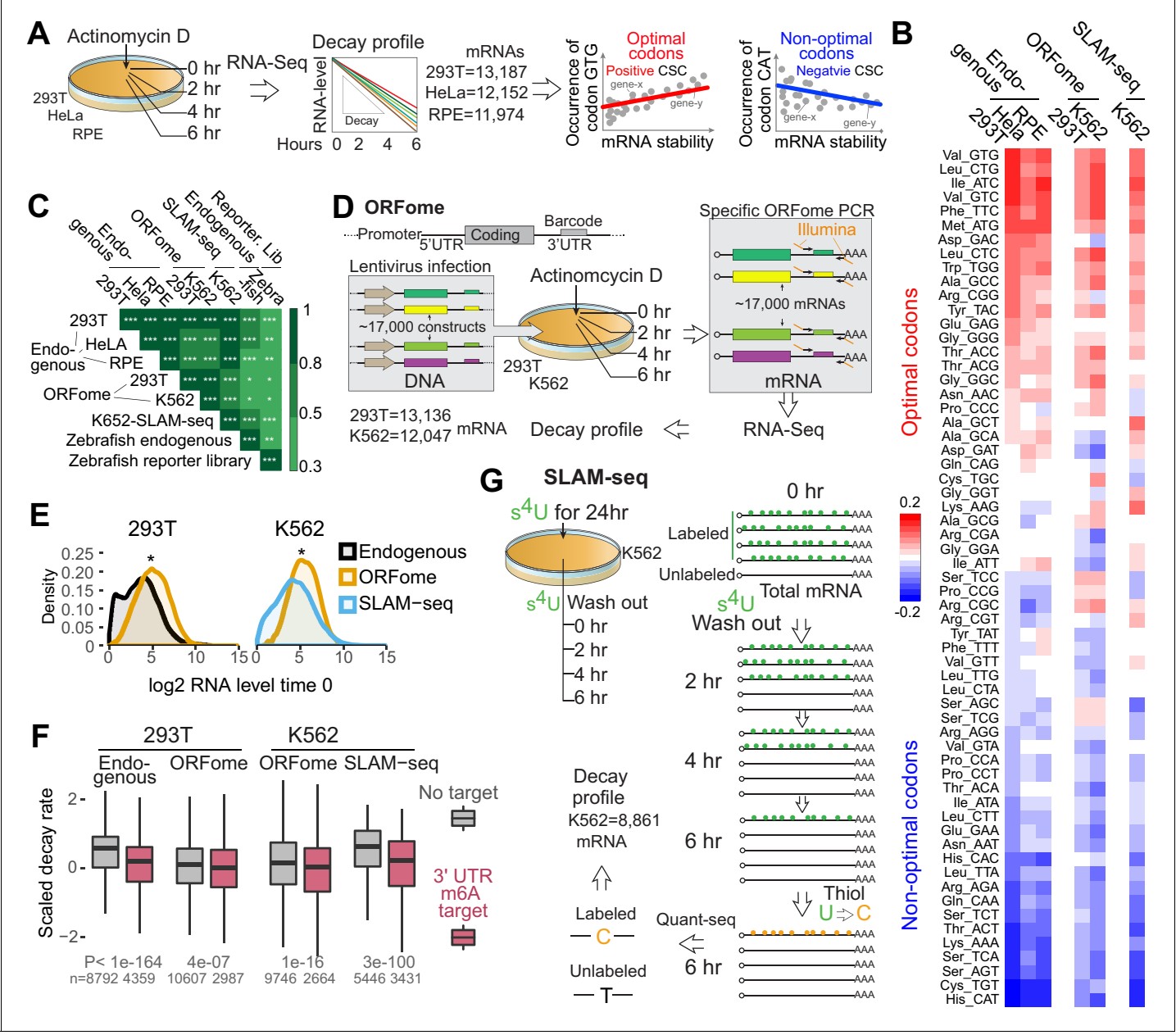

**Figure 1.** Human mRNA decay correlate with codon composition in human lines. (**A**) Scheme of the mRNA endogenous decay profiles approach in 293T, HeLa and RPE cells. RNA-seq was performed at different time after blocking transcription by Actinomycin D treatment. The codon stabilization coefficient (CSC) is calculated as the Pearson correlation coefficient between the occurrence of each codon and mRNA stability. (**B**) Heatmap showing the codon stability coefficient (CSC) calculated in the indicated cell lines and approaches used to measure mRNA stability. (**C**) Heatmap showing the Spearman rank correlations between CSC calculated in different human cells lines and using different methods to calculate mRNA stability. CSC calculated in zebrafish embryos with two independent methods were included (**Bazzini et al., 2016**). *p<0.05, **p<0.01 and ***p<1e-16. (**D**) Diagram illustrating the ORFome decay profiles approach. Cells were infected with lentivirus carrying approximately 17,000 expression vectors containing different coding sequences but sharing the same promoter, 5' and 3'UTRs. Each vector contains a 20-nucleotide long unique barcode in the 3'UTR. Infected cells were selected by puromycin treatments. RNA was isolated at different time after blocking transcription by Actinomycin D. Specific RNA-seq libraries were generated amplifying the shared sequences surrounding the barcodes and mRNA ORFome decay was calculated based on the sequences. (**E**) Density distribution of the mRNA level in 293T or K562 cells using the indicated techniques. Two-sample Kolmogorov-Smirnov test is used, *p<1e-50. (**F**) Boxplot showing the scaled mRNA decay of previously defined m6A targets mRNAs and as control, mRNAs that were not defined at m6A targets in the indicated cell line and using the indicated method to calculate decay. A linear model is used to estimate the difference. Number of genes (n) and p value KS test indicated. (**G**) Schematic representation of the SLAM-seq. Cells were fed with s4U every 3 hr for 24hs hours.

*Figure 1 continued on next page*

*Figure 1 continued*

Orthogonal-chemistry-based RNA sequencing was preformed after washing out the s4U. Decay profiles were calculated based on the chase of the label/unlabeled mRNAs (T to C conversion).

DOI: https://doi.org/10.7554/eLife.45396.003

The following source data and figure supplement are available for figure 1:

**Source data 1.** mRNA stability profiles used in *Figure 1*.
DOI: https://doi.org/10.7554/eLife.45396.005
**Source data 2.** CSC scores from *Figure 1B*.
DOI: https://doi.org/10.7554/eLife.45396.006
**Figure supplement 1.** Human and mouse mRNA decay correlate with codon composition in human lines.
DOI: https://doi.org/10.7554/eLife.45396.004

measures endogenous mRNA half-lives based on 4-thiouridine (s$^4$U) incorporation and orthogonal-chemistry-based RNA sequencing, where incorporated s$^4$U is read as cytosine (C) rather than uracil (U) (*Herzog et al., 2017*). Cells were 'fed' with s$^4$U, followed by washout and unlabeled chase (*Figure 1G* and *Figure 1—figure supplement 1D, E, F, G* and methods). We observed a strong CSC score correlation using the SLAM-seq approach akin to the exogenous ORFome (*Figure 1B,C*) approach as well as sequencing of endogenous mRNAs in different cell types (*Figure 1B,C* and *Figure 1—figure supplement 1B*). Interestingly, we also observed strong half-lives correlation between our SLAM-seq data and a similar methodology, TimeLapse-seq (*Schofield et al., 2018*) in K562 cell (*Figure 1—figure supplement 1H*) and so, strong CSC scores correlation (*Figure 1—figure supplement 1I*). Furthermore, SLAM-seq has also been done in mouse embryonic stem cells (*Herzog et al., 2017*), we calculated CSC scores based on their data, it also correlated with our human SLAM-seq data (*Figure 1—figure supplement 1I*), indicating a similar codon optimality code in mouse embryonic stem cells. And similar to our endogenous mRNA, mRNAs previously defined as targets of the m6A pathway display increased decay compared to a control set of mRNAs in our SLAM-seq (*Figure 1F*). In sum, we have measured mRNA decay using three independent approaches in different human cells (293T, HeLa, RPE and K562) to score codon-optimality-code. Our results suggest that codon composition affects mRNA stability in all the tested human cell lines, and that the regulatory properties of each codon are similar between cells.

## Codon identity affects mRNA stability in human cell lines

The results above indicate that regulatory information is embedded in the coding sequence. To determine whether this sequence information is 'read' in a codon-dependent manner or simply nucleotide composition, we compared a pair of reporters that differed by a single nucleotide insertion (1nt frame-shift), causing a frameshift that converts an 'optimal' coding sequence (enriched in optimal codons) into a 'non-optimal' sequence (enriched in non-optimal codons) (*Figure 2A*), keeping the nucleotide composition otherwise almost identical (*Figure 2A*). We redeployed a reporter system where mCherry (red fluorescent protein) or GFP (green fluorescent protein) was followed by a ribosome skipping sequence (P2A) (*de Felipe et al., 2006*; *Donnelly et al., 2001*) and a coding region enriched in either optimal or non-optimal codons (due to 1 nucleotide frameshift) (*Figure 2A*). The P2A sequence allowed the analysis of protein production in vivo independent of potential folding effects from the optimal or non-optimal peptide. If the regulatory information is 'read' in a manner that was dependent on codon identity, there should be a correlation between codon-optimality scores and expression levels. We found that mRNA derived from the non-optimal reporter was less stable than mRNA derived from its optimal counterpart after blocking transcription with Actinomycin D (*Figure 2B*). Further, this difference was observed at RNA level 24 hr post-transfection (*Figure 2C*), suggesting that codon composition can affect homeostatic mRNA levels (*Bazzini et al., 2016*). In addition to higher mRNA abundance, we also observed higher fluorescence intensity from the optimal reporter vs the non-optimal reporter (*Figure 2D*). No significant differences were observed in the co-transfection control (GFP), ruling out potential global expression changes due to toxicity of the non-optimal peptide (*Figure 2D*). These changes due to the codon composition were independent of the nucleotide used to cause the frameshift (G, C, U or A) (*Figure 2—figure supplement 1A*), fluorescence protein used in the reporter assay (mCherry-based, *Figure 2D*; GFP-based reporters, *Figure 2—figure supplement 1B*) or coding sequence of the

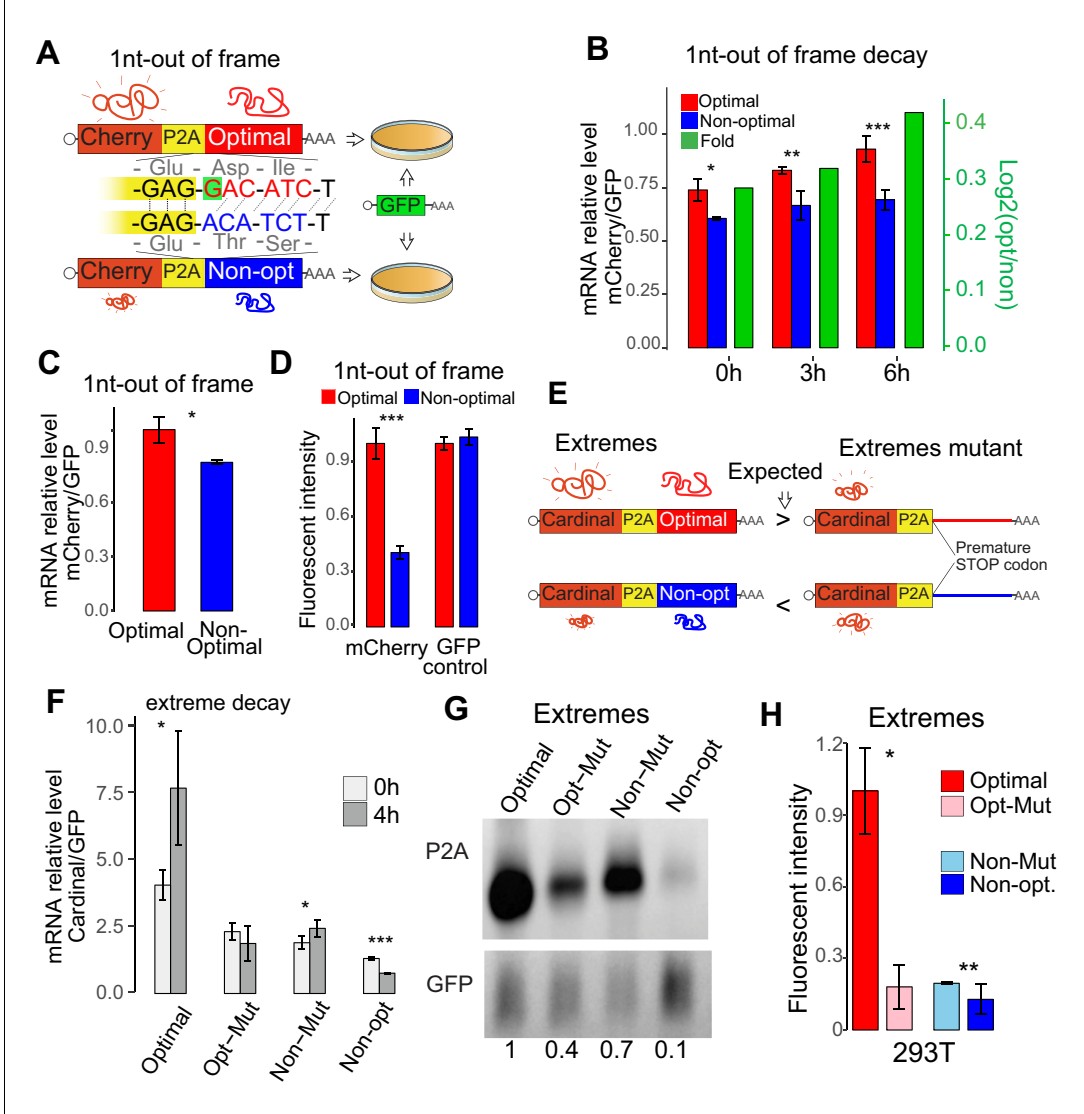

**Figure 2.** Codon identity affects mRNA stability in human lines. (**A**) Scheme of the 1nt-out of frame reporters: two mRNAs that differ in the codon composition due to a single nucleotide (G in red, highlight with green) which creates a frameshift. The encoding mCherry fluorescent protein was followed by a cis-acting hydrolase elements (P2A) and then by the coding region enriched in optimal or non-optimal codons due to the frame shift. P2A causes ribosome skipping, therefore, the mCherry is not fused to the optimal or non-optimal encoded proteins. The reporter pairs were co-transfected with a vector encoding for GFP as internal control. (**B**) The 1nt-out of frame reporter enriched in optimal codons displayed higher stability than its non-optimal counterpart. Cells are treated with Actinomycin D 24hs after transfection, samples are collected at different timepoint to measure RNA level after blocking transcription, qPCR result is normalized with GFP internal control. *p<0.05, **p<0.01 and ***p<0.005 Unpaired t.test for all the panels. (**C**) qRT-PCR results showing that the 1nt-out of frame reporter enriched in optimal codons present higher level of RNA than its non-optimal counterpart 24 hr post-transfection. (**D**) The 1nt-out of frame reporter enriched in optimal codons displayed higher mCherry intensity than its non-optimal counterpart measure by cytometry. None significant differences were observed in the internal GFP control. (**E**) Scheme of the 'Extreme' reporters. Pairs of mRNA differing only in single mutation causing a premature stop codon. Each mRNAs contain Cardinal fluorescent protein coding region followed by a cis-acting hydrolase elements (P2A) and coding region enriched in optimal or non-optimal codons. Their counterparts (referred as mutants) contain a single mutation that cause a premature stop codon upstream the region enriched in optimal or non-optimal codons. Therefore, the translational competent reporter enriched in optimal codons should evidence higher level of expression that its mutant counterpart. And the reporter in non-optimal mRNA should express less than its counterpart translational incompetent. (**F**) qRT-PCR results showing relative stability of extreme reporters. Cells are treated with Actinomycin D 24hs after transfection, samples are collected at different timepoint to measure RNA level after blocking transcription, GFP is used as internal control to normalize qPCR result. The extreme translation-competent reporter enriched in optimal codons presented higher stability than its translation deficient counterpart (Mut). And the translation-competent reporter enriched in non-optimal codon was more unstable than its translation deficient counterpart (Mut). (**G**) Northern-blot showing that the extreme reporter enriched in optimal codons presented higher RNA level that its mutant counterpart. Contrary, the mRNA enriched in non-optimal codons present lower mRNA level than is mutant counterpart. GFP mRNA signal was used as internal control. Relative normalized quantifications are indicated. (**H**) The extreme reporter enriched in optimal codons displayed

*Figure 2 continued on next page*

*Figure 2 continued*

higher fluorescent intensity than its mutant counterpart measure by cytometry. The reporter enriched in non-optimal codons evidenced lower fluorescent intensity than its mutated pair.

DOI: https://doi.org/10.7554/eLife.45396.007

The following source data and figure supplement are available for figure 2:

**Source data 1.** Reporter sequences and oligos used for *Figure 2*.

DOI: https://doi.org/10.7554/eLife.45396.009

**Figure supplement 1.** Codon identity affects mRNA stability in human lines in a translation-dependent manner.

DOI: https://doi.org/10.7554/eLife.45396.008

frame-shifted reporter (different frame-shifted coding sequence pair, *Figure 2—figure supplement 1C*). Likewise, similar outcomes were observed in HeLa, RPE and K562 cells (*Figure 2—figure supplement 1D*), consistent with strong correlation between their respective codon optimality scores (*Figure 1*). Taken together these results indicate that regulatory information is dependent on codon assignment rather than other nucleotide sequence information.

To assess whether active translation in cis is required to confer the regulatory effects on stability, we generated different paired reporters (Extremes) with single nucleotide mutation to create a premature stop codon, preventing the region enriched in either optimal or non-optimal to be translated (*Figure 2E*). Differently to the 1nt-out of frame reporters (*Figure 2A*), the 'extreme' reporters enriched in optimal or non-optimal codons do not share the same nucleotide composition but contain higher optimality differences. Therefore, if translation in cis is important, translation of a coding region enriched in either optimal or non-optimal codons should increase or decrease mRNA stability, respectively, when compared to their untranslated counterparts (referred to as Mut) (*Figure 2E*). Indeed, a translation-competent reporter enriched in optimal codons resulted in, increased mRNA stability (*Figure 2F*), mRNA abundance (*Figure 2G*) and fluorescent intensity (*Figure 2H*) compared to its translation-deficient counterpart in 293 T cells. Conversely transfection of a translation-competent reporter enriched in non-optimal codons resulted in decreased mRNA stability, mRNA abundance, as well as fluorescence intensity, when compared to its translation-deficient counterpart in 293 T cells (*Figure 2F,G,H*). Similar fluorescent intensity differences were observed in HeLa cells (*Figure 2—figure supplement 1E*). Furthermore, codon-mediated effects on endogenous mRNAs were dampened in both directions upon inhibition of translation through cycloheximide treatment. Specifically, the absolute correlation coefficient between mRNA stability and codon occurrence, CSC, were smaller in 293 T cells treated with cycloheximide (*Figure 2—figure supplement 1F*), when compared to untreated 293 T cells (*Figure 1*). In the presence of cycloheximide, mRNAs enriched in optimal codons were less stable and mRNAs enriched in non-optimal were less unstable when compared to untreated cells (*Figure 2—figure supplement 1G*). This data demonstrates that mRNA stability and expression are influenced by the codon composition in a translation-dependent manner.

## Synonymous codons affect mRNA stability in human cell lines

The above results indicate that codon identity impacts mRNA stability in human cells. In yeast, translation elongation speed is related to codon optimality, where 'non-optimal' codons are more slowly translated compared to 'optimal' codons. The decoding rate can be influenced by both amino acid identity (*Artieri and Fraser, 2014*; *Gardin et al., 2014*) and tRNA levels. To address whether amino acid identity influences mRNA stability, we first assessed whether synonymous codons behaved in the same way. We hypothesized that synonymous codons should share the same optimality behavior if amino acid identity was the main determinant. For example, in zebrafish and *Xenopus*, we found that some amino acids share either optimal or non-optimal codons (*Figure 3A*) (*Bazzini et al., 2016*). Of the 20 different amino acids, we found that in human, only histidine possessed synonymous codons that were exclusively non-optimal across all three assays (*Figure 3A*). For a few other amino acids (e.g. glycine, serine), synonymous codons were largely encoded by optimal or non-optimal codons in most of the assays (*Figure 3A*). For the majority of amino acids, synonymous codons possessed different regulatory properties. However, a preprint recently proposed that amino acid identity also acts as a driver of translation-dependent decay regulation using similar dataset in

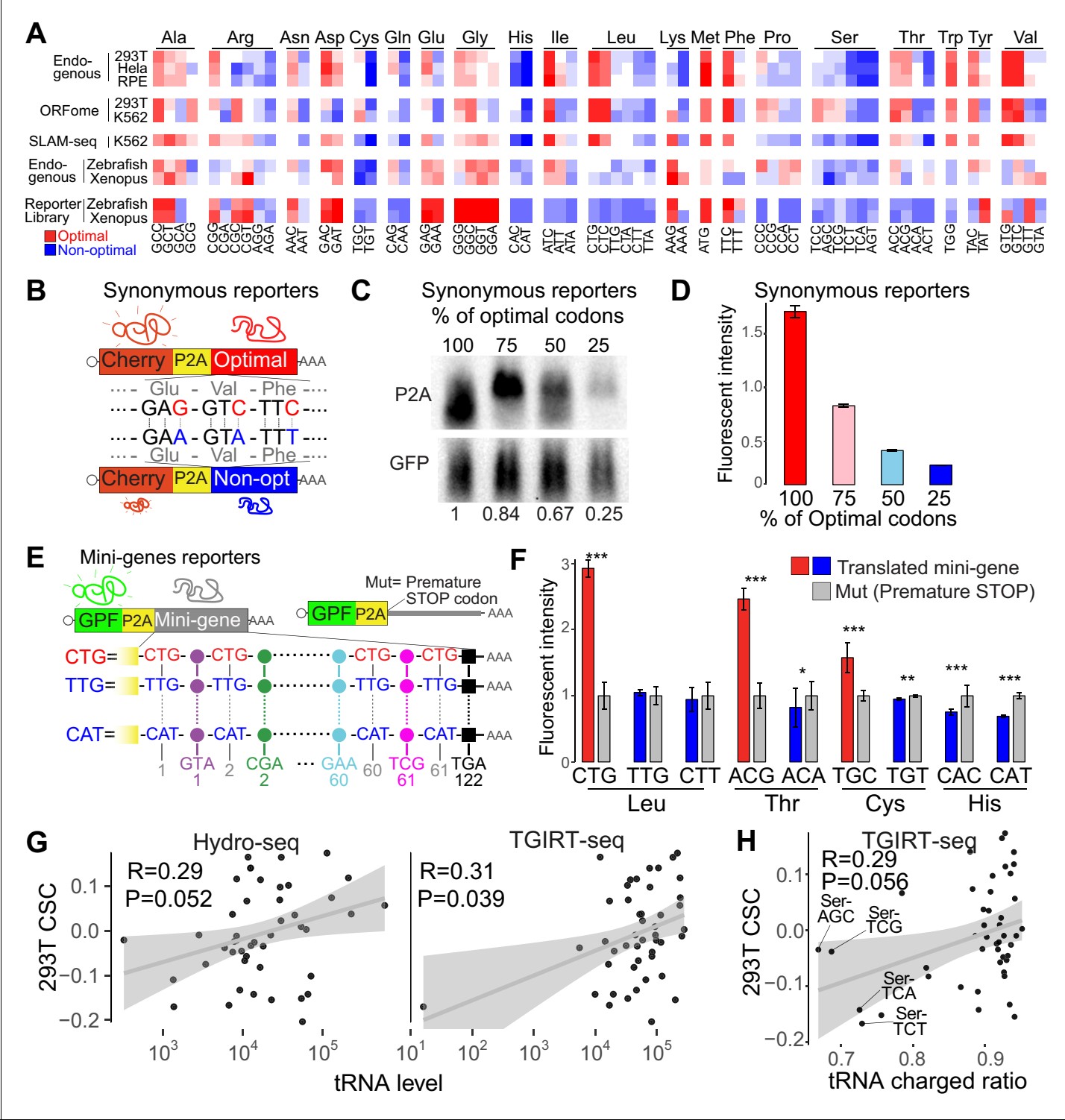

**Figure 3.** Synonymous codons affect mRNA stability in human lines. (A) Heatmap showing the codon stability coefficient (CSC) sorted by the encoded amino-acid, for the indicated cell lines and approaches used to measure mRNA stability. The codon optimality scores calculated in zebrafish and *Xenopus* are also shown (***Bazzini et al., 2016***). (B) Diagram of the 'Synonymous' reporters. mRNAs differing only in synonymous mutations with different regulatory effects on mRNA stability. Each mRNA contains the coding region for mCherry fluorescent protein followed by a *cis*-acting hydrolase elements (P2A) and a coding region that different in the proportion of optimal and non-optimal codons but encoding for the same peptide (synonymous mutations). (C) Northern-blot showing that the synonymous reporter enriched in more optimal codons presented higher RNA level. GFP was used as internal control. Relative normalized quantifications are indicated. (D) The fluorescent intensity correlated with the proportion of optimal

*Figure 3 continued on next page*

*Figure 3 continued*

codon of each of the four reporter mRNAs differing only in synonymous codons. (E) Scheme of 'Mini-gene' reporters: all mini-genes mRNAs contain the coding region for Green fluorescent protein followed by a *cis*-acting hydrolase elements (P2A) and a coding region that contain the 61 different coding codons (colored circles) alternated with a single codon that give name to the mini-gene. The counterpart pair of each mini-gene (Mut) contains a single mutation in the P2A that prevent the coding region enriched in one particular codon to be translated. (F) Fluorescent GFP intensity of the indicated mini-genes normalized with internal control (mCherry). Unpaired t.test is used *p<0.05, **p<0.01, ***p<0.005. (G) Scatter plots showing the 293T CSC (endogenous mRNA) and tRNA levels quantified by Hydro-seq (*Gogakos et al., 2017*) and TGIRT-seq (*Evans et al., 2017*), Spearman rank correlations indicated. (H) Scatter plots showing the 293T CSC (endogenous mRNA) and tRNA charged ratios quantified by TGIRT-seq with periodate oxidation (*Evans et al., 2017*). The four nuclear tRNAs encoding for Serine are indicated. Spearman rank correlations indicated.
DOI: https://doi.org/10.7554/eLife.45396.010

The following source data and figure supplement are available for figure 3:

**Source data 1.** Reporter sequences and oligos used for *Figure 3*.
DOI: https://doi.org/10.7554/eLife.45396.012
**Figure supplement 1.** Few amino acids can potentially be related with mRNA stability in human lines.
DOI: https://doi.org/10.7554/eLife.45396.011

human (*Forrest et al., 2018*), which shows strong correlation with ours CSC scores (*Figure 3—figure supplement 1A*). Therefore, we calculated the amino acid stabilization coefficient (ASC), as the Person correlation coefficient between mRNA stability and amino acid occurrence (*Bazzini et al., 2016*). Similar to the preprint (*Forrest et al., 2018*), amino acids presented correlations with mRNA decay (ASC) (*Figure 3—figure supplement 1B*). However, in human those correlations are strongly affected after removing the most abundance synonymous codon for most of the amino acids (*Figure 3—figure supplement 1B*). For example, Leucine and Isoleucine displayed a positive ASC score suggesting that both amino acids might be optimal (also mentioned in *Forrest et al., 2018*); however, in all the datasets both amino acids are encoded by optimal and non-optimal codons and the most abundance codon dominates the ASC calculation (*Figure 3—figure supplement 1B*). Actually, both amino acids would be referred as non-optimal after removing the single most abundant synonymous codon (*Figure 3—figure supplement 1B*). However, as mentioned before, there are few amino acids like Histidine or Serine (*Figure 3—figure supplement 1B*), where all the synonymous codons temp to share the same optimality trend (*Figure 3A* and *Figure 3—figure supplement 1B*), and so those few amino acids might have an implication on mRNA stability.

To further test that codons, and less likely the amino acid they encode, affect mRNA stability, we sought to determine the effect of synonymous codons in mRNA stability using two different approaches. First, using reporter genes differing in silent mutations (*Figure 3B*), we observed that mRNA levels (*Figure 3C*) and fluorescent intensity (*Figure 3D*) were higher in the reporters enriched in optimal codons compared to counterparts enriched in non-optimal codons. These results suggested that synonymous codons can have different effects on gene expression. Second, we designed individual reporters (Mini-gene) that contained coding sequences, in which all even positions (61 of 122 codons) after the P2A releasing sequence possessed a single codon type (*Figure 3E*). As before, we measured expression levels between translation-competent and translation-deficient reporters (due to premature stop codon, Mut) (*Figure 3E*). Due to the enrichment on a single codon in each mini-gene reporter, the Mini-gene reporters provide a way to analyze the regulatory properties at a single codon resolution. Consistent with our observations of endogenous mRNAs, we found that several amino acids possessed synonymous codons that differed in optimality identity (*Figure 3F*). Specifically, mini-genes enriched in optimal codons displayed increased expression when compared to translation-deficient counterparts, whereas mini-genes enriched in non-optimal displayed either decreased expression, or no significant change when compared to respective translation-deficient counterparts (*Figure 3F*). These observations further validate the differences in optimality between synonymous codons for leucine, cysteine and isoleucine (*Figure 3F* and *Figure 3—figure supplement 1B*). As predicted, in the case of Histidine (*Figure 3A* and *Figure 3—figure supplement 1B*), both mini-gene reporters presented the same non-optimality trend found for endogenous profiles (*Figure 3F*) (with CAT being consistently more non-optimal than CAC, (*Figure 3A*)). These several results together suggest that codon identity affects mRNA stability in a translation-dependent manner.

To explore how synonymous codons display different regulatory behaviors, we examined their respective tRNA levels, which have been proposed to be an important determinant of codon optimality (*Bazzini et al., 2016*; *Presnyak et al., 2015*). tRNA levels measured by two independent techniques in 293T cells, Hydro-tRNA-seq and TGIRT-seq (*Evans et al., 2017*; *Gogakos et al., 2017*), correlates with 293T CSCs (*Figure 3G*), supporting the idea that tRNA levels affect mRNA stability. However, these correlations are not strong, and the two methods do not strongly correlate ((*Figure 3G* and *Figure 3—figure supplement 1C*) , evidencing that measuring tRNA level is challenging. Beside the level of total tRNA, we investigate the relation between tRNA quality and codon optimality. Interestingly, the ratio of charged tRNA (with amino acid) to total tRNAs (with and without amino acid) in 293T cells also correlated with the CSC (*Figure 3H*), suggesting that the ratio of charged tRNA might also affect mRNA stability. In particular, the four encoded nuclear tRNAs for Serine were the most uncharged tRNAs and Serine is one of the amino acids that appears to be non-optimal in human, zebrafish and *Xenopus* (*Figure 3A*) (*Bazzini et al., 2016*). This suggests that particular amino acids may contribute to codon-mediated mRNA decay (*Bazzini et al., 2016*; *Frumkin et al., 2017*). Our results suggest that both tRNA level and quality contribute to the available 'tRNA ready to go' (*Rak et al., 2018*), that might dictate the codon-mediated regulation of mRNA stability.

## Poly(A)-tail is affected but not required for the codon optimality mechanism in human and zebrafish

Poly(A)-tail length and alternative polyadenylation influences the translation and stability of mRNAs (*Geisberg et al., 2014*; *Lima et al., 2017*; *Moqtaderi et al., 2018*; *Subtelny et al., 2014*; *Tian and Manley, 2013*; *Tian and Manley, 2017*). Consistent with this, in zebrafish and yeast endogenous and reporter mRNAs enriched in optimal codons tend to possess longer poly(A)-tails than mRNAs enriched in non-optimal codons (*Bazzini et al., 2016*; *Mishima and Tomari, 2016*; *Radhakrishnan et al., 2016*; *Webster et al., 2018*). In the present study, two independent observations indicate that in humans, mRNAs enriched in optimal codons tend to have longer poly(A)-tails than genes enriched in non-optimal codons. First, we calculated polyadenylation status by comparing levels of poly(A+)-containing mRNA against total RNA depleted of ribosomal RNA. Similar to zebrafish, genes enriched in optimal codons presented higher polyadenylation status (poly(A)/total RNA ratio) compared to those enriched in non-optimal codons in RPE cells (*Figure 4A*). Second, we analyzed two independent datasets measuring the poly(A)-tail length in a transcriptome-wide manner, PAL-seq (*Subtelny et al., 2014*) and TAIL-seq (*Chang et al., 2014*), in 293T and HeLa cells, respectively. In both datasets, mRNAs enriched in optimal codons displayed longer poly(A)-tails than mRNAs enriched in non-optimal codons (*Figure 4B*). Together, these results indicate that in human cells poly(A)-tail length correlates with codon composition within mRNAs.

Although poly(A)-tails length corelates with codon-optimality and stability (*Figure 4A,B*) (*Bazzini et al., 2016*; *Mishima and Tomari, 2016*; *Radhakrishnan et al., 2016*; *Webster et al., 2018*), no causal relationship between them has been established. Therefore, we next set out to determine whether codon effects on stability are mediated through the modulation of poly(A)-tail length. Toward this end, we generated reporter genes that possess a histone tail (*Figure 4C*), with unique 3' end structure found on canonical non-polyadenylated histone genes (*Marzluff et al., 2008*). The histone-tailed reporter mRNAs displayed precise 3' end (*Figure 4C, D* and *Figure 4—figure supplement 1B*). Despite invariable length and the absence of a poly(A)-tail, histone-tailed reporter mRNAs enriched in optimal genes displayed higher RNA levels (*Figure 4D*) and protein intensity (*Figure 4E* and *Figure 4—figure supplement 1B*) compared to counterparts enriched in non-optimal codons. Further, we wondered whether this dissociation between poly(A)-tail and codon optimality is specific to human or a more general phenomenon, so we injected the reporters into zebrafish embryos. Injected mRNAs into zebrafish embryos, possessing either a cytoplasmic poly(A) signal or a histone tail displayed similar trends of fluorescence intensity change based on codon composition (*Figure 4C, F* and *Figure 4—figure supplement 1C, D*). These results demonstrate that modulation in poly(A)-tail length is not required for codon-mediated changes in mRNA stability in human and zebrafish embryos. This implies that poly(A)-tail shortening in non-optimal mRNAs likely occurs in parallel to or as a consequence of mRNA destabilization rather than as the primary cause of codon-mediated destabilization.

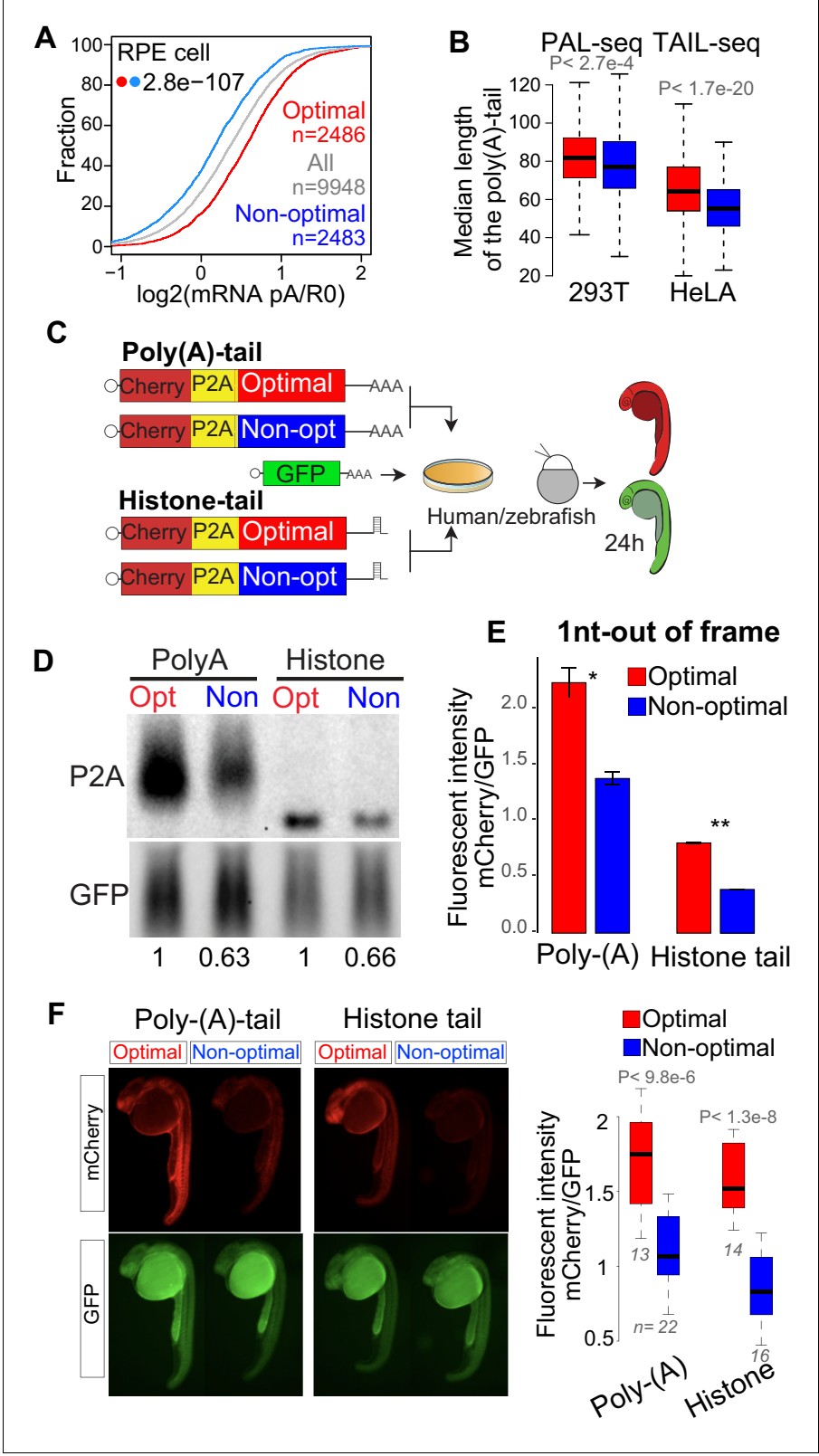

**Figure 4.** Poly(A)-tail is affected but not required for the codon optimality mechanism in human and zebrafish. (**A**) Cumulative distribution of the poly(A) selected mRNA and total mRNA ratio in RPE cells showing that mRNA enriched in optimal codons present higher poly(A) ration than mRNA enriched in non-optimal codons, p-*value* indicated, Wilcoxon rank-sum test. (**B**) Box plots showing the length of the poly(A)-tails of mRNA enriched in
*Figure 4 continued on next page*

*Figure 4 continued*

optimal or non-optimal codons measured by PAL-seq (*Subtelny et al., 2014*) or TAIL-seq (*Chang et al., 2014*), the top 500 genes enriched in optimal or codons genes are used for each group, p-*value* indicated, Wilcoxon rank-sum test. (**C**) The 1nt-out of frame reporter containing either poly(A)- or histone-tail were transfected into human cells or co-injected into single-cell stage zebrafish embryos with a GFP construct as internal control. (**D**) Northern-blot showing that the 1nt-out of frame reporters enriched in optimal codons presents higher level of mRNA comparing to the counterpart enriched in non-optimal codon containing either a poly(A)- or histone- tail signal at 24 hr post-transfection. GFP was used as internal control. Relative normalized quantifications are indicated. (**E**) The 1nt-out of frame reporters enriched in optimal codons displayed higher fluorescent intensity than it counterparts enriched in non-optimal codons measure by cytometry, independent of the poly(A)- or histone-tail used. Unpaired t.test is used. *p<0.01, **p<0.005. (**F**) Fluorescence microscopy images of representative embryos at 24 hr post injection with the indicated 1nt-out of frame reporter and GFP. Box plot displays fluorescence quantification of 24 hr embryos injected with each reporter. mCherry fluorescence intensity was normalized to GFP intensity in each embryo, Poly-(A), n > 13 pairs, p = 9.3e-06 and Histone, n > 14 pairs, p = 1.3e-08, Wilcoxon rank-sum test.

DOI: https://doi.org/10.7554/eLife.45396.013

The following source data and figure supplement are available for figure 4:

**Source data 1.** Reporter sequences and oligos used for *Figure 4*.

DOI: https://doi.org/10.7554/eLife.45396.015

**Figure supplement 1.** Poly(A)-tail is not required for the codon optimality mechanism in human and zebrafish.

DOI: https://doi.org/10.7554/eLife.45396.014

## The number of ribosomes translating a mRNA modulates the codon-mediated effects on gene expression

Codon-mediated regulation of mRNA stability depends on translation in human cells (*Figure 2F, G, H* and *Figure 2—figure supplement 1E, F, G*) as well as in other model organisms (*Bazzini et al., 2016*; *Mishima and Tomari, 2016*; *Presnyak et al., 2015*). Our data and published studies indicate that translational elongation, which may be influenced by tRNA abundance, is correlated with stability kinetics. Based on these data, we hypothesized that the ribosome is a key regulatory molecular factor, therefore, we proposed that increasing the number of ribosomes in a mRNA population would enhance the codon-meditated effects on gene expression. To explore this hypothesis, we generated paired reporters (1nt frame-shift) possessing a battery of different 5' and 3' UTR sequences from mRNAs with different levels of translation based on zebrafish ribosome profiling (*Figure 5A*) (*Bazzini et al., 2014*). The ability of these sequences to affect protein production was confirmed in human cells (*Figure 5B* and *Figure 5—figure supplement 1A*). We also generated reporters possessing upstream open-reading frame (uORF) elements within the 5'UTR, known to repress translation of the canonical reading frame (*Figure 5A* and *Figure 5—figure supplement 1C*) (*Johnstone et al., 2016*). In all cases, we found that the impact of codon composition on mRNA stability was enhanced in reporters that displayed a higher baseline rate of translation. Specifically, optimal and non-optimal paired reporters with high translation rates displayed greater differences at both RNA and protein levels when compared to similarly paired reporters with lower translation efficiencies (*Figure 5B, C-* and *Figure 5—figure supplement 1A, B, C*). These results indicate that the level of translation, and therefore, the number of ribosomes loaded onto an mRNA, influences codon-mediated effects on gene expression.

Since the number of ribosomes on mRNA can modulate codon-mediated effects on gene expression, we hypothesized that *trans*-regulatory elements and physiological conditions where mRNA translation is globally affected, may also impact the codon-mediated effects on gene expression. Viruses are known to reduce endogenous mRNA translation (*Walsh et al., 2013*). For example, Herpes simplex virus 1 (HSV-1) infection reduces the translation efficiency of endogenous mRNAs globally (*Figure 5—figure supplement 1D*) (*Rutkowski et al., 2015*). Therefore, we compared homeostatic levels of endogenous mRNAs before and after HSV-1 infection. We observed that genes enriched in optimal codons presented higher mRNA level than genes enriched in non-optimal codons at homeostasis in all cell lines analyzed (*Figure 5D* and *Figure 5—figure supplement 1E*). Interestingly, the differences in mRNA levels between genes enriched in optimal versus non-optimal codons were dampened after infection (*Figure 5D*). In agreement with our reporter results

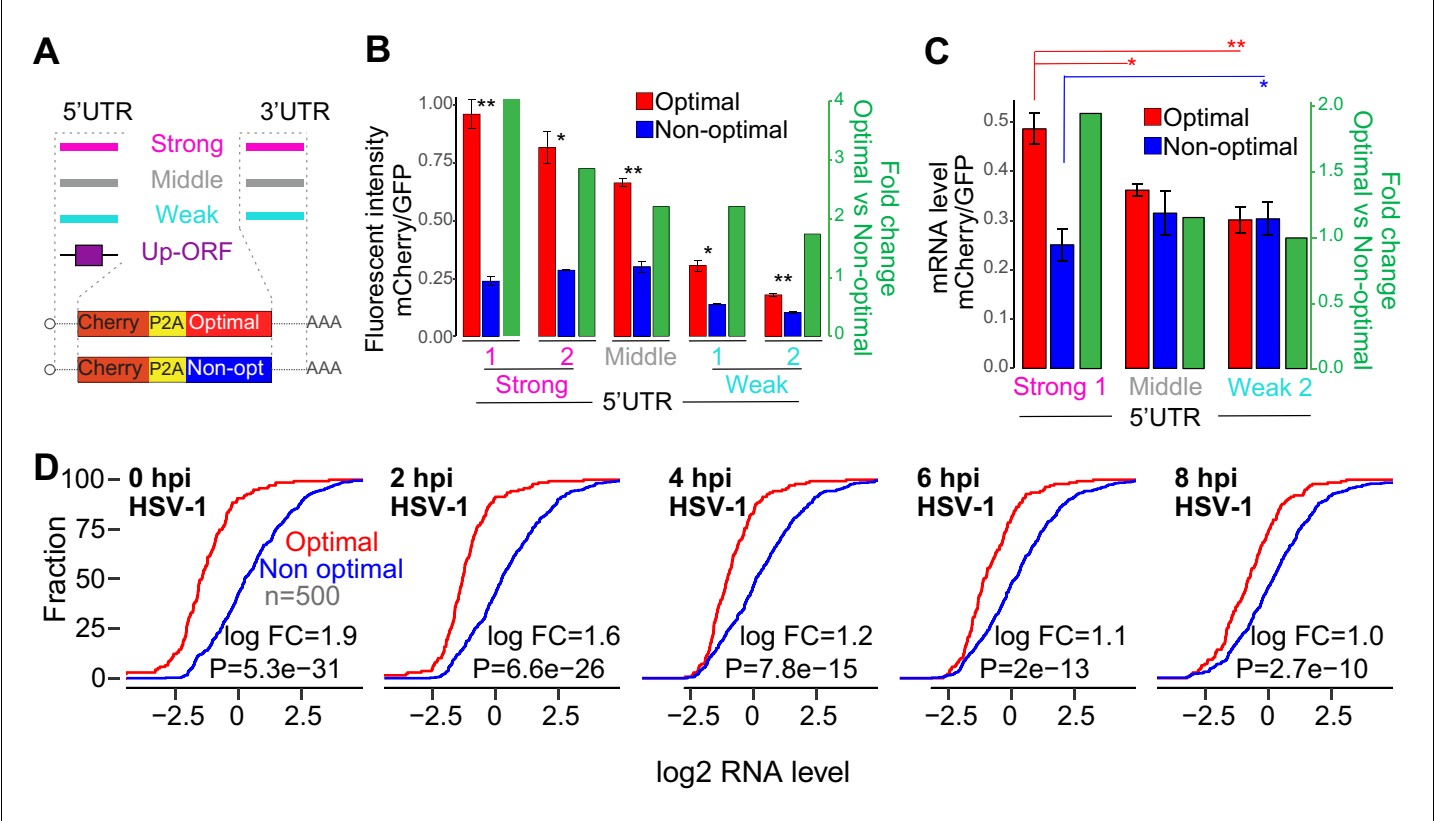

**Figure 5.** The number of ribosomes translating a mRNA modulates the codon-mediated effects on gene expression. (**A**) Schematic representation of '1nt-out of frame' reporters using different 5'UTR or 3'UTR or upstream ORFs (Up-ORF) driving different level of translation. (**B**) The fluorescent intensity correlated with the nature of the 5'UTR used as well as the fold change of fluorescent intensity between the reporter enriched in optimal and non-optimal codons. Unpaired t.test is used, *p < 1e-3, **p < 1e-4. (**C**) Reporters enriched in optimal codons displayed higher level of RNA when 5'UTR driving higher level of translation were used compared to constructs using 5'UTR driving lower level of translation. Contrary, the reporter enriched in non-optimal codons using the 5'UTR driving higher level of translation presented lower level of RNA than the one containing the weakest 5'UTR. Unpaired t.test is used, *p<0.05, **p<0.01. (**D**) Cumulative distribution of mRNA level from mRNAs enriched in either optimal or non-optimal codons during HSV-1 virus infection, p-*value* indicated, Wilcoxon rank-sum test.

DOI: https://doi.org/10.7554/eLife.45396.016

The following source data and figure supplement are available for figure 5:

**Source data 1.** Reporter sequences and oligos used for *Figure 5*.
DOI: https://doi.org/10.7554/eLife.45396.018
**Figure supplement 1.** The number of ribosomes translating a mRNA modifies the codon-mediated effects on gene expression.
DOI: https://doi.org/10.7554/eLife.45396.017

(*Figure 5C*), when translation was reduced during virus infection, genes enriched in optimal codons were downregulated and mRNA enriched in non-optimal codons upregulated compared to control set of genes (*Figure 5—figure supplement 1F*). These results suggest that the impact of codon optimality on gene expression is conditional upon other *cis-* (e.g. UTRs) and *trans-* (e.g. virus) regulatory mechanisms within the cell that modulate translation level.

## Discussion

The main function of the ribosome is to translate the mRNA nucleotide sequences into the amino acid sequence. However, translation is also important for mRNA quality control, targeting defective mRNAs for degradation (*Shoemaker and Green, 2012*). Translation has also been shown to strongly affect mRNA stability of non-defective transcripts depending on the codon composition in model organisms (*Bazzini et al., 2016*; *Boël et al., 2016*; *Burow et al., 2018*; *de Freitas Nascimento*

*et al., 2018*; *Harigaya and Parker, 2016*; *Jeacock et al., 2018*; *Mishima and Tomari, 2016*; *Presnyak et al., 2015*; *Radhakrishnan et al., 2016*). Here, we demonstrate that, in human cells (also suggested in *Hanson and Coller, 2018*), translation also strongly affects mRNA stability in a codon-dependent manner, thus dictating mRNA and protein levels at homeostasis (*Figure 6*). These observations highlight the wealth of regulatory information residing within the coding sequence, the largest fraction of the human transcriptome. Further, this study provides insight on the regulatory role of core components, such as tRNAs and ribosomes (*Figure 6*). These have long been under appreciated in regulation despite their relatively high levels of expression, their abundant interactions with mRNAs, and evolutionary conservation. Future studies of this relatively unexplored mechanism of post-transcriptional regulation may be relevant to human diseases.

Here, we used diverse approaches to measure mRNA decay of both exogenous and endogenous mRNA in different human cell lines. We observed that particular codons were enriched within stable mRNAs and other codons were enriched within unstable mRNAs, optimal and non-optimal codons, respectively (*Figure 1*). In particular, we developed a method (ORFome) to distinguish mRNA decay regulation between regulation coming from codon composition versus other regulatory information (UTRs). In addition to highlighting codon effects, this method proved to be a viable strategy to identify other *cis* regulatory features impacting stability. For example, by comparing endogenous decay profiles with ORFome profiles, we found that predicted m6A targets displayed considerable different destabilization (*Figure 1F*).

We demonstrated that the regulatory information was based on codon identity rather than other sequence information, by using reporter gene pairs differing in the codon composition due to a

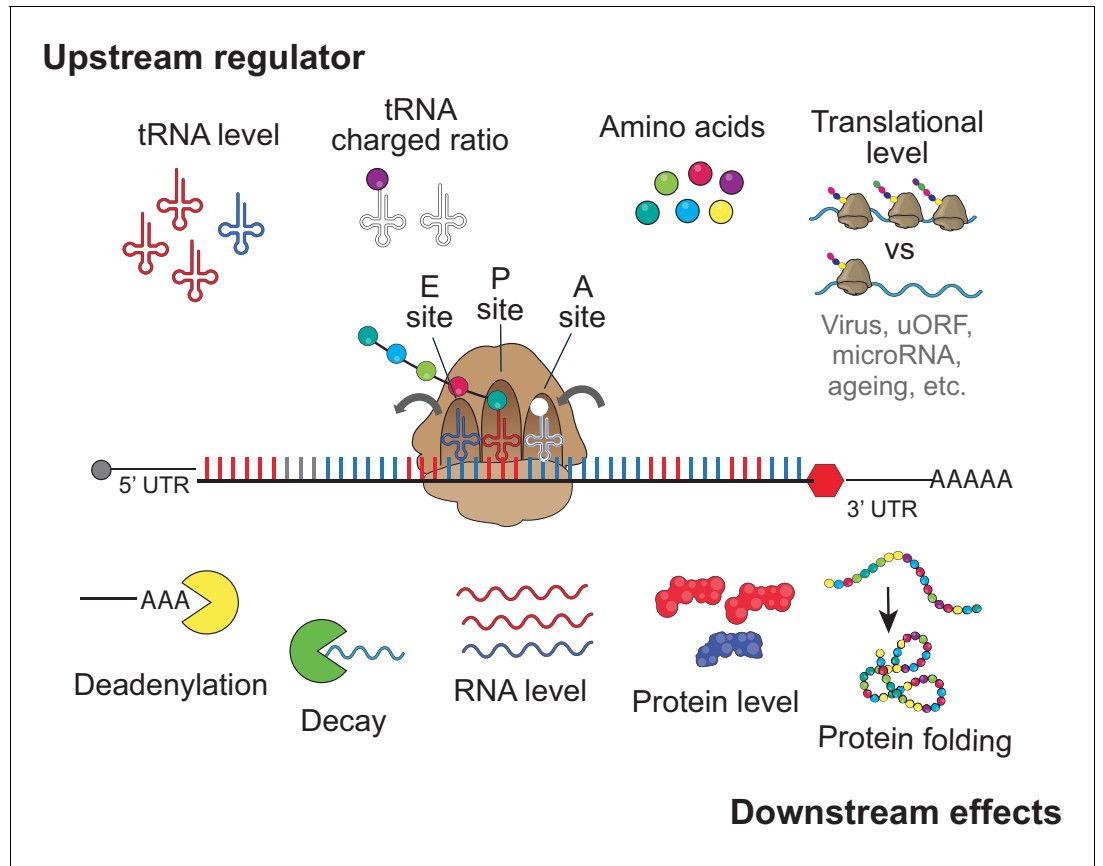

**Figure 6.** upstream regulator and downstream effect for codon optimality tRNA level, tRNA charged ratio, amino acid, and translational level might contribute to regulates the regulatory identity and/or strength of each codon to affect gene expression, by influencing the speed of translation elongation. The downstream effects of the codon-meditated mechanism are RNA deadenylation, mRNA decay, mRNA level, protein level and protein folding.

DOI: https://doi.org/10.7554/eLife.45396.019

single nucleotide insertion causing a framed shift (*Figure 2*). While most synonymous codons affect mRNA stability differently, a few amino acids (e.g. Histidine, Serine) (*Figure 3*), appear to have an effect at the amino acid level by either impacting the rate of peptidyl transfer or the amino-acyl tRNA charging step (potentially, Serine) (*Figure 6*). Comparing to human, in zebrafish and *Xenopus* embryos (*Bazzini et al., 2016*), few more amino acids could be proposed to be optimal or non-optimal, suggesting potential multilayer regulatory differences between species or cell stage (embryogenesis). In human cells, a variety of data indicate that codons rather than the amino acid they encode—affects mRNA stability in a translation-dependent manner. First, most amino acids (e.g Leucine, Isoleucine, Cysteine, Glutamic acid, Arginine, Asparagine, Phenylalanine, Threonine, Tyrosine, Valine) display both optimal and non-optimal codons in most decay profiles (*Figure 3*). Second, two set of reporters having synonymous mutations confirmed that synonymous codon affects mRNA stability differently, especially the mini-genes we designed have high resolution at single codon level (*Figure 3*). Third, we observed, that non-optimal codons tend to be decoded by lowly expressed tRNAs, and optimal codons by highly expressed tRNAs (*Figure 3*). This suggests that tRNA supply/ demand might be an important determinant of codon-mediated mRNA stability (*Figure 6*) (*Bazzini et al., 2016*; *Despic and Neugebauer, 2018*; *Frumkin et al., 2018*; *Richter and Coller, 2015*). Furthermore, we observed that non-optimal codons also displayed a low ratio of charged to total tRNA (*Figure 3*). Interestingly, in the case of serine, defined as non-optimal, all nuclear-encoded tRNAs were the most poorly charged, suggesting that charging tRNA may be another crucial point impacting codon-mediated gene regulation (*Figure 6*). Therefore, the 'tRNA ready to go' level (*Rak et al., 2018*) may be modulating translation elongation rates and serving as a determinant of codon optimality (*Figure 6*). In addition of aminoacyl charging, tRNA quality is also related to tRNA modifications, therefore, it will be worthwhile to examine the effects of tRNA modifications on mRNA stability (*Chou et al., 2017*; *Duechler et al., 2016*). Our data support the idea that tRNAs are master gene regulators (*Gingold et al., 2014*; *Goodarzi et al., 2016*) and highlight the need to fully understand how tRNA expression and processing are regulated in different cells and different conditions, including human diseases (*Ishimura et al., 2014*; *Kirchner and Ignatova, 2015*; *Torrent et al., 2018*). Therefore, genes involve in tRNA expression regulation and processing, would be strong candidates to explain a part of the codon optimality molecular mechanism.

With respect to the function of the ribosome in mRNA quality control, it was proposed that codon optimality might be a Slowness-mediated decay (SMD) (*Rak et al., 2018*) because it is similar to a slow-go-decay (SGD) pathway where the elongation speed of the ribosome might be the trigger for regulation. This is distinct from no-go-decay (NGD), where stalling ribosomes trigger mRNA degradation (*Simms et al., 2017*). Therefore, codon optimality may be related to mRNA quality control mechanisms for: recognition, labeling and cleaning of the mRNA. However, the molecular mechanisms of how translation affects mRNA stability in a codon-dependent manner remains poorly characterized. While Dhh1p in yeast senses ribosome elongation speed (*Radhakrishnan et al., 2016*) and affects mRNA stability in a codon-dependent manner, the role of the vertebrate ortholog, DDX6, with respect of codon optimality is not clear. In human cells, there is a clear correlation between mRNA enriched in non-optimal codons and shortening of the poly(A)-tail (*Figure 4*) similar to that observed in yeast and zebrafish (*Bazzini et al., 2016*; *Mishima and Tomari, 2016*; *Radhakrishnan et al., 2016*; *Webster et al., 2018*). However, reporter genes possessing a histone tail and therefore lacking a poly(A)-tail were still affected by codon composition in both human and zebrafish embryos (*Figure 4*). Therefore, these results indicate that the poly(A)-tail is not required and likely, the shortening of the poly(A)-tail in genes enriched in non-optimal codons is an indirect consequence of decreased stability rather than a required step in the codon optimality mechanism (*Figure 6*). Future work should explore the molecular mechanism of how the translation affects mRNA destabilization.

In addition to altered dynamics of mRNA stability due to specific misregulation of tRNA levels (*Goodarzi et al., 2016*; *Kirchner and Ignatova, 2015*; *Torrent et al., 2018*), we showed that codon-mediated mRNA stability can be affected by altering the translational level through either global changes in translation efficiency (during viral infection) or specific 5' and 3' UTRs sequences (*Figure 5*). This suggests an alternative way for the cell to differentially regulate gene expression (*Figure 6*). Therefore, it will be important to integrate the evolutionary and functional relationship between translation initiation driven by the UTRs and translation elongation dictated by the coding sequence with respect to codon-mediated gene regulation (*Chu et al., 2014*) as well as the effects

of codon composition and translation speed on protein folding (*Yu et al., 2015*) (*Figure 6*). The observation that codon optimality effects can be modulated by altering baseline translation (*Figure 5*), leads us to speculate that, global reductions in translation efficiency might explain the predicted attenuated codon optimality effect in neural-specific mRNA decay in *Drosophila* (*Burow et al., 2018*). Therefore, molecular factors such microRNAs or RNA-binding protein regulating mRNA translation could potentially modulate the impact of codon-mediated regulation (*Figure 6*). Hence, it is worth exploring how codon-mediated regulation is impacted in disease states where translation rates are globally affected, such as in neurodegenerative disease (*Gao et al., 2017*), ribosomopathies (*Danilova and Gazda, 2015*), virus infection (*Walsh et al., 2013*) and other cellular stresses (*Gonskikh and Polacek, 2017*). In addition to the cell-specific translational competency (ribosome specificity, translation level), the ready to go tRNA repertoire (level, charged) and intrinsic mRNA properties (mRNA localization, codons distribution along the coding region) need to be integrated to understand the differential gene expression during development, cell reprogramming, as well as identify underlying causes of misregulated genes in human diseases.

# Materials and methods

**Key resources table**

| Reagent type | Designation | Source or reference | Identifiers | Additional information |
|---|---|---|---|---|
| Cell line (*Homo sapiens*) | HEK293T | American Type Culture Collection | #CRL-11268 | |
| Cell line (*H. sapiens*) | Hela | American Type Culture Collection | #CCL2 | |
| Cell line (*H. sapiens*) | K562 | American Type Culture Collection | #CRL-4000 | |
| Cell line (*H. sapiens*) | RPE | American Type Culture Collection | #CCL-243, Lot #62995945 | |
| Recombinant DNA reagent | ORFome library | Sigma-Aldrich | TRC3 ORF Puromycin Arrayed Glycerol Library | |
| Commercial assay or kit | North2South Hybridization and Detection Kit | ThermoFisher | Cat#17097 | |
| Commercial assay or kit | SLAMseq Kinetics Kit | Lexogen | Cat#061.24 | |
| Software, algorithm | R for analysis and plot | version 3.5.1 | | |

## Tissue culture

293T, HeLa and RPE cells were cultured with DMEM media, supplied with 10% FBS. K562 cells were cultured with IMDM media with 10% FBS. The cells were ordered from tissue culture facility from the Stowers Institute, at relative low passage, lower than passage 15.

## Transfection

293T, HeLa and RPE cells were transfected with lipofectamine 3000 based on manufacture's instruction in 24-well plate. K562 cell was transfected with Trans-X2 (from Mirus company) in 96-well plate. 24 hr post transfection, cells are collected for cytometry or RNA extract.

## Cytometry analysis

The florescent reporter intensity of the cells was quantified in ZE5 equipment using GFP (488/510) and mCherry (587/610), cells were suspended in DMEM with 10%FBS. Cells were not fixed. Cytometry data was analyzed with FlowJo, median intensity of the cells was used to represent fluorescent intensity.

## RNA decay

Cells were sub-cultured in 24-well plate overnight, so the cells were at around 80% coverage before treatment. Actinomycin D was added into the well, with final concentration of (5 µg/ml), in 0.1% DMSO. Cycloheximide was added at 2 µg/ml, in $H_2O$. Cells were directly collected with Trizol, at desired time-point for RNA extract.

## RNA extract and qPCR

RNA was extracted using Trizol reagent and quantified with Qubit RNA broad range kit for RNA-seq or RT and qPCR. For reverse transcription, superscript IV kit was used from Invitrogen. qPCR was done using Perfect SYBR Green FastMix Reaction Mixes, QuantaBio.

## Northern-blot

RNA gel running was following the protocol from Lonza Bioscience. Shortly. Total RNA was resuspended with 1xMOPS buffer, formaldehyde and deionized formamide. heat at 70℃ for 10 min, chill on ice, and add loading buffer before running. Then RNA was migrated using 1X MOPS at 100V for 3 hr and transferred with 10X SSC overnight. Oligonucleotide DNA probes with 3'biotin were ordered from IDT with HPLC purification. Probing and detection were done following the protocol of North2South Chemiluminescent Hybridization and Detection Kit from ThermoFisher, using Streptavidin-HRP.

## ORFome infection

The ORFome library was bought from Sigma-Aldrich company named:TRC3 ORF Puromycin Arrayed Glycerol Library, the commercial version already contains specific barcode in each mRNA. The 96 well format ORFome library was pool together using an automatic robot. First in 20 bins were created based in ORF size, then those 20 were pooled in five groups, grown, DNA maxi-preparation done. After quantification, the five groups were mixed into a single cocktail. The ORFome cocktail was transfected with lentivirus vector: Pax2 and VSV-G into 293 T cells for virus packing. Media containing virus were filtered and ultracentrifuged to enrich virus for infection. 293 T cells and k562 cells were infected and selected with puromycin for a week, at the concentration of 293T(2 µg/ml), K562 (0.75 µg/ml). Then, cells were treated with Actinomycin D (5 µg/ml) at six-well plate and samples were collected in duplicate every hour 0–6 hr for RNA-seq. Specific oligos were used to target the surrounding barcode region of ORFome to generate library and sequenced, we tested different PCR cycles to amply the library and finally used 15 cycles. The ORFome reads were trimmed using cutadapt 1.16; the resulting trimmed reads were mapped to the ORF-ome barcodes using salmon 0.9.1. To calculate mRNA stability, transcripts were selected with the Bioconductor zFPKM package, a cut-off of zFPKM > −3 was applied. The decay rate was estimated using a linear model according to the integrated rate law for a first-order reaction.

## SLAM-seq

SLAMseq Kinetics Kit - Catabolic Kinetics Module) from Lexogen was performed. Basically, k562 cells were feed with 100 uM 4sU for 24 hr, fresh media containing s4U was changed every 3 hr. For chasing, old media are removed and cells were washed with PBS three times before adding media without s4U, but 10 mM UTP instead, samples were collected at 0, 2, 4, 6 hr in triplicates for RNA extract and IAA treatment. All the operations were done in red light source to protect 4sU from crosslink. QuantSeq 3'mRNA-seq library were generated and sequenced, by PCR 12 cycles for library amplification based on the protocol. The SLAM-seq data was processed with the slam dunk pipeline (https://github.com/t-neumann/slamdunk 0.3.3.) The reads were aligned to the human genome GRCh38. The half-life was estimated according to the SLAM-seq paper; for downstream analyses a cut-off of p-value<0.05 was applied to the decay rate estimates.

Seven set of evidences indicated that we have successfully performed SLAM-seq and Quant-seq (*Herzog et al., 2017*). (1) As expected, the majority of sequenced reads come from the 3'UTR. (2) From all possible nucleotide transitions across the transcriptome, T-to-C changes were only significant in cells treated with s4U (*Figure 1—figure supplement 1D*). (3) The rate of T-to-C transitions decrease with time after washing out the s4U from the medium, consistent with labeled ('old') mRNA being replaced with new mRNA (unlabeled) (*Figure 1—figure supplement 1E*). (4) We observed a strong correlation between mRNA levels calculated with regular RNA-seq (CPM) and Quant-seq (length-independent) as well as between technical replicates (*Figure 1—figure supplement 1F*). (5) The s4U incorporation does not affect global mRNA abundance when compared to untreated embryos (*Figure 1—figure supplement 1G*). (6) The mRNA half-life we get from SLAM-seq highly correlates with the published mRNA half life from TimeLapse-seq (*Figure 1—figure supplement 1H*). (7) As shown in *Figure 1G*, m6A mRNA targets presented lowered stability than a control mRNA group.

## Zebrafish injection

The plasmid constructs containing optimal and non-optimal ORFs with poly A tail and Histone tail were linearized with Not1HF or Kpn1HF respectively, similarly GFP containing control plasmid was also linearized with Not1HF. Linearized plasmids were then in vitro transcribed using SP6 mMessage mMachine kit (Life technology). The mRNA with polyA tail(150 ng/μl), GFP(100 ng/μl) and mRNA with histone tail (200 ng/μl) plus GFP(100 ng/μl) were microinjected separately for each construct in zebrafish embryos at one-cell stage. Injected embryos were imaged after 24 hr using 'Zeiss Lumar. V12 steREO' microscope with same conditions for all the injected embryos. Raw images were processed using Fiji software separately for poly(A)-tail and histone-tail constructs.

## 3'RACE assay for histone tail

The end of histone mRNA is detected by 3'RACE. Total RNA is ligated with the adaptor:/5rApp/ AGATCGGAAGAGCACACGTCTGAACTCCAGTCAC/3ddC/by T4 Rnl2tr K227Q. cDNA was produced with following oligo: GTGACTGGAGTTCAGACGTGTGCTCTTCCG, And PCR with specific forward oligo and the RT oligo. PCR product was inserted into TOPO vector, and miniprep for sanger sequencing.

All the vector sequences, primers and probes can be accessed from the Stowers Original Data Repository at http://www.stowers.org/research/publications/libpb-1373.

## Acknowledgements

We thank C Takacs (University of New Haven), K Si and R Krumlauf (Stowers Institute) for suggestions and critical reading of the manuscript. We also thank A Zemouri and E Viode for help during their short training in our lab. We also thank the following Stowers Core facilities: Aquatics, Molecular Biology, Cytometry, Screening, Tissue Culture and Microscopy. Specifically, B Rubinstein, K Zhang, M Miller, K Weaver, K Hall, A Perera and S Chen for extraordinary support. This study was supported by the Stowers Institute for Medical Research. This work was performed as part of thesis research for QW, Graduate School of the Stowers Institute for Medical Research. Original data underlying this manuscript can be accessed from the Stowers Original Data Repository at http://www.stowers.org/research/publications/libpb-1373.

## Additional information

### Funding

| Funder | Author |
| --- | --- |
| Stowers Institute for Medical Research | Ariel Alejandro Bazzini |

The funders had no role in study design, data collection and interpretation, or the decision to submit the work for publication.

## Author contributions
Qiushuang Wu, Conceptualization, Formal analysis, Validation, Investigation, Methodology, Writing—original draft, Project administration, Writing—review and editing; Santiago Gerardo Medina, Data curation, Formal analysis, Methodology; Gopal Kushawah, Matthew Wright, Data curation, Validation; Michelle Lynn DeVore, Luciana A Castellano, Validation, Methodology; Jacqelyn M Hand, Validation; Ariel Alejandro Bazzini, Conceptualization, Supervision, Funding acquisition, Investigation, Methodology, Writing—original draft, Project administration, Writing—review and editing

## Author ORCIDs
Qiushuang Wu (iD) http://orcid.org/0000-0002-9301-3630
Ariel Alejandro Bazzini (iD) https://orcid.org/0000-0002-2251-5174

## Ethics
Animal experimentation: This study was performed in strict accordance with the recommendations in the Guide for the Care and Use of Laboratory Animals of the National Institutes of Health. All of the zebrafish were handled according to approved institutional animal care and use committee (IACUC) protocols (#2016-0159) of the Stowers Institute for Medical Research. The protocol was approved on August 18th 2018.

## Decision letter and Author response
Decision letter https://doi.org/10.7554/eLife.45396.033
Author response https://doi.org/10.7554/eLife.45396.034

## Data availability
The following datasets have been uploaded to NCBI GSE, under the accession number GSE126523. Endogenous mRNA decay: 1. 293T: 0h, 0h, 1h, 2h,3h,6h,6h 2. Hela: 0h,0h,1h,2h,3h,4h,5h,6h,6h 3. RPE: 0h,0h,1h,2h,3h,4h,5h,6h,6h ORFome: 1. 293t: 0h, 0h, 1h, 1h, 2h, 2h, 3h, 3h, 4h, 4h, 5h, 5h, 6h, 6h 2. K562: 0h, 0h, 1h, 1h, 2h, 2h, 3h, 3h, 4h, 4h, 5h, 5h, 6h, 6h SLAM-seq: 1. K562: 0h, 0h, 0h, 2h, 2h, 2h, 4h, 4h, 4h, 6h, 6h, 6h

The following dataset was generated:

| Author(s) | Year | Dataset title | Dataset URL | Database and Identifier |
|---|---|---|---|---|
| Shiyuan (Cynthia) Chen | 2019 | Three different approaches to measure endogenous and exogenous mRNA decay in four human cell lines (293T, Hela, RPE and K562) | https://www.ncbi.nlm.nih.gov/geo/query/acc.cgi?acc=GSE126523 | NCBI Gene Expression Omnibus, GSE126523 |

The following previously published datasets were used:

| Author(s) | Year | Dataset title | Dataset URL | Database and Identifier |
|---|---|---|---|---|
| Rutkowski AJ, Erhard F, L'Hernault A, Bonfert T, Schilhabel M, Crump C, Rosenstiel P, Efstathiou S, Zimmer R, Friedel CC | 2015 | HSV1 infection data | https://www.ncbi.nlm.nih.gov/geo/query/acc.cgi?acc=GSE59717 | NCBI Gene Expression Omnibus, GSE59717 |
| Gogakos T, Brown M, Garzia A, Meyer C, Hafner M, Tuschl T | 2017 | hydro- tRNA seq, table S1 | https://www.ncbi.nlm.nih.gov/geo/query/acc.cgi?acc=GSE95683 | NCBI Gene Expression Omnibus, GSE95683 |
| Evans ME, Clark WC, Zheng G, Pan T | 2017 | TGIRT tRNA seq and charged ratio | https://www.ncbi.nlm.nih.gov/geo/query/acc.cgi?acc=GSE97259 | NCBI Gene Expression Omnibus, GSE97259 |
| Chang H | 2014 | TAIL-seq hela | https://www.ncbi.nlm.nih.gov/geo/query/acc. | NCBI Gene Expression Omnibus, |

| | | cgi?acc=GSE54114 | GSE54114 |
|---|---|---|---|
| Subtelny AO | 2014 PAL-seq 293T | https://www.ncbi.nlm.nih.gov/geo/query/acc.cgi?acc=GSE52809 | NCBI Gene Expression Omnibus, GSE52809 |

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
