## [Decision Letter]

Thank you for submitting your article "Translation affects mRNA stability in a codon dependent manner in human cells" for consideration by *eLife*. Your article has been reviewed by two peer reviewers, and the evaluation has been overseen by a Reviewing Editor and Kevin Struhl as the Senior Editor. The following individuals involved in the review of your submission have agreed to reveal their identity: Jonathan S Weissman (Reviewer #1).

The reviewers have discussed the reviews with one another and the Reviewing Editor has drafted this decision to help you prepare a revised submission.

Summary:

Wu et al. investigated the contribution of codon optimality and related properties to the stability of transcripts in human cells. They conclude that the speed of ribosome elongation along a transcript inversely correlates the susceptibility of that transcript to decay and degradation – and that a major contributor to ribosome speed is codon identity (and corresponding tRNA abundance). This agrees with relationships between codon optimality and mRNA degradation that have been observed in many other organisms. The question is important as it has not been not completely resolved.

Essential revisions:

The observation that half-life and codon usage are related in multiple cell types and using multiple measurements of half-life is a valuable contribution, and is based on a large number of experiments. These datasets also are useful for future studies.

As you see from the comments of reviewer #2, a bioRxiv manuscript appeared that reported very similar results to yours. Because the interpretation of the very similar data is different, we are asking you to discuss the bioRxiv manuscript.

In light of the reviewers' comments and Forrest's paper, you should also reanalyze your data. You ought to more strongly support the connection between CSC and half-life beyond tRNA abundance since that correlation is weak, and that perhaps you could perform some of the analyses in Forrest et al. paper to test alternative models for the CSC-half-life connection.

Your conclusion is expected but has not been shown before and therefore might have turned out differently. Because if this we believe that the paper should be published as "Tools and Resources" paper?

I include the full reviews of the referees because they are very thoughtful and informative.

*Reviewer #1:*

A connection between the codon composition of an mRNA's coding sequence and the half-life of the corresponding transcript has recently been observed in several organisms, including yeast, zebrafish, and *Drosophila*. Similar effects appear to not yet have been published in mammalian cells (a 2018 review (PMID: 29018283) claims that such an effect has been observed in "human, mouse and rat cell lines" but cites only unpublished observation). Using a variety of experimental approaches, Wu et al. demonstrate that codon content and mRNA stability are correlated in several human cell lines.

The authors globally measure the half-life of mRNAs by performing a time course of RNAseq after shutting off new transcription with actinomycin D treatment. They produce qualitatively similar measurements without incurring potential secondary effects from global transcriptional shutoff using a recently developed metabolic labeling approach. They also measure half-lives of a synthetic library of transcripts in which a large set of endogenous ORFs are inserted between the same 5' and 3' UTRs. Calculating the correlation between the frequency with which each codon occurs in a transcript and that transcript's measured half-life in each experimental approach allows the authors to quantify the effect of that codon on mRNA stability (the codon stability coefficient, or CSC). They observe largely consistent CSC values for each codon across 293T, HeLa, RPE, and K5262 cells and across the three approaches used.

To confirm that transcript stability is directly influenced by translation of different codon types (i.e. that this correlation does not simply reflect co-evolution between codon usage and other transcript features influencing stability), the authors follow up by measuring the relatively stability of various reporters in which coding sequences enriched or depleted in "optimal" codons have been placed downstream of fluorescent reporter, separated by a P2A skipping sequence. This allows mRNA abundance to be read out directly by qPCR or indirectly by abundance of the fluorescent protein. Their results support the claim that translation of particular codons can directly stabilize or destabilize transcript. They also demonstrate a positive but weak correlation between CSC and the abundance of each codon's cognate tRNAs from recently published sequencing-based measurements.

Major points:

The most striking feature of the original observation that codon identity affected transcript stability in yeast was that stabilizing codons consisted almost entirely of codons that had been independently annotated as optimal based on genomic tRNA copy number and codon-anticodon base pairing. While the concept of codon optimality is more complex in human than in yeast, the manuscript is light on analysis of the features that codons observed to stabilize or destabilize transcripts have in common. Understanding such features would help to evaluate different models for how stabilization/destabilization is mechanistically achieved. While the correlations between CSC and tRNA abundance/charged ratio in Figure 3G and H are a step in this direction, these results seem overstated in the text. The authors claim that "our results suggest that the 'tRNA ready to go' availability (Rak et al., 2018) largely determines codon-mediated regulation on mRNA stability" (emphasis ours) and that "non-optimal codons are decoded by lowly expressed tRNAs, and optimal codon by highly expressed tRNAs". Neither of these statements are well supported by the data, in which it seems clear that tRNA abundance explains at most a small fraction of the variance in CSC and that there are many low-tRNA-abundance codons with high CSC and vice versa. Interpretation of Figure 3G and H would be helped by a scatter plot showing how consistent the Hydro-seq and TGIRT-seq estimates of tRNA abundance are with each other. (The fact that the extreme outlier point at 10^1 in the TGIRT-seq scatter plot is not an outlier in the Hydro-seq data suggests at least some inconsistency). How strong are Pearson (as opposed to the reported Spearman rank) correlations for 3G and H? (The use of Spearman is reasonable since there is no reason to expect a linear relationship, but the data does not visually suggest a coherent non-linear but monotonic relationship.) How do measured CSC values compare to codon optimality as estimated by relative frequency of usage in the transcriptome or in a subset of highly expressed housekeeping genes?

Global measurements of mRNA half-lives by metabolic labeling and sequencing have recently been published in K562s (PMID 29355846) and in mouse embryonic stem cells (PMID 28945705). The manuscript would also be strengthened by both confirming that CSC values inferred from the published K562 data agree with the author's data and by evaluating whether the effect is observed in mouse cells as well.

*Reviewer #2:*

In this study, Wu et al. attempt to decipher the contribution of codon optimality and related properties to the stability of transcripts in human cells. The basic thesis is that speed of ribosome elongation along a transcript inversely correlates with (and perhaps mediates?) the susceptibility of that transcript to decay and degradation – and that a major contributor to ribosome speed is codon identity (and corresponding tRNA abundance). This is consistent with relationships between codon optimality and mRNA degradation that have been observed in many other systems.

To support their model, the authors generate datasets measuring RNA stability using three orthogonal sequencing methods, including sequencing post transcriptional shutdown (via ActD), sequencing after a s4U pulse-chase (SLAM-seq), and sequencing of a pool of human ORFs in a lentiviral system with (nearly) invariant 5' and 3' UTRs. Their observations broadly support the hypothesis that codon identity contributes to transcript stability. This has been reported previously, both in yeast (Presnyak et al., 2015) and in human cells (Forrest et al., 2018), but the authors' comprehensive datasets make this a valuable study. In addition, the array of reporters that the authors have constructed (including the synonymous/non-syn; 1 nt frameshift and the mini genes) are valuable tools to study the contributions of codons towards translation, and the results support the broad thrust of the authors work. Their observation that codon dependent effects are not dependent on and possibly causal for or orthogonal to polyA tail length is also intriguing.

Major concern:

There is an almost identical dataset and analysis that was recently reported, which is not cited or discussed (Forrest et al., 2018). Forrest et al. perform almost identical analysis to Wu et al. Figure 3G, H using the same preexisting data – but come to opposite conclusions. Wu et al. conclude that based on Figure 3G, H with correlations of ~0.3 that "tRNA levels.… correlates with 293T CSCs", while Forrest et al. conclude based on correlations of ~0.3 are "not nearly as robust as that seen in yeast; thus, other aspects of codon identity could be at play." Forrest et al. goes on to demonstrate that the driver of this transcript instability is the amino acid identity, that there are certain amino acids (mostly hydrophobic ones) that are inefficient in forming peptide bonds, and this (along with the tRNA abundance and codon optimality) leads to the alterations in ribosomal speed – but Wu et al. conclude that amino acids play no role. It seems to this reviewer that it is important for Wu et al. to discuss the conclusions in Forrest et al., especially when the same data are used to derive different conclusions. Forrest et al. is currently a preprint but it is freely available and the conclusions are highly related – sometimes even involving the same previously published datasets – so it should be discussed.

Other concerns:

1) It is this reviewer's understanding that the ORFeome library used by the authors was generated with invariant 5' and 3' UTRs (Yang et al., 2011). However, the authors indicate that the library has unique barcodes in the 3' UTR, which enables them to perform specific PCR across the barcodes, sidestepping the problem of distinguishing the ORfeome derived reads from endogenous genes during sequencing. This reviewer was unable to find a Materials and methods section describing this barcoded library (if the authors have generated one, and a reference if not). The Materials and methods section for sequencing library prep should be elaborated upon.

2) Furthermore, the authors refer to the ORFeome library as having "common promoter, 5' and 3' UTRs" but if the 3' UTRs have ORF-specific barcodes then they are not common to all library members. How can the authors be sure that including barcodes in the 3' UTR does not affect mRNA stability?

3) It is unclear why the authors attribute the narrow distribution in mRNA levels of the ORFeome to sharing a strong promoter as opposed to having fewer regulatory elements in their untranslated regions – or some combination thereof.

4) The approaches used to generate sequencing libraries in this work (QuantSeq and SLAM-seq and the ORFeome barcoding sequencing) all generate libraries with much lower diversity than standard RNA-seq (since they just sequence the very 3' end of molecules or a barcode region). Because of this, PCR duplicates become more of an issue. How many cycles of PCR were performed? How were PCR duplicates handled?

Forrest, Megan E., Ashrut Narula, Thomas J Sweet, Daniel Arango, Gavin Hanson, James Ellis, Shalini Oberdoerffer, Jeff Coller, and Olivia S Rissland. 2018. 'Codon usage and amino acid identity are major determinants of mRNA stability in humans', bioRxiv.

Presnyak, V., N. Alhusaini, Y. H. Chen, S. Martin, N. Morris, N. Kline, S. Olson, D. Weinberg, K. E. Baker, B. R. Graveley, and J. Coller. 2015. 'Codon optimality is a major determinant of mRNA stability', Cell, 160: 1111-24.

Yang, X., J. S. Boehm, X. Yang, K. Salehi-Ashtiani, T. Hao, Y. Shen, R. Lubonja, S. R. Thomas, O. Alkan, T. Bhimdi, T. M. Green, C. M. Johannessen, S. J. Silver, C. Nguyen, R. R. Murray, H. Hieronymus, D. Balcha, C. Fan, C. Lin, L. Ghamsari, M. Vidal, W. C. Hahn, D. E. Hill, and D. E. Root. 2011. 'A public genome-scale lentiviral expression library of human ORFs', Nat Methods, 8: 659-61.

---

## [Author Response]

Essential revisions:The observation that half-life and codon usage are related in multiple cell types and using multiple measurements of half-life is a valuable contribution, and is based on a large number of experiments. These datasets also are useful for future studies.As you see from the comments of reviewer #2, a bioRxiv manuscript appeared that reported very similar results to yours. Because the interpretation of the very similar data is different, we are asking you to discuss the bioRxiv manuscript.In light of the reviewers' comments and Forrest's paper, you should also reanalyze your data. You ought to more strongly support the connection between CSC and half-life beyond tRNA abundance since that correlation is weak, and that perhaps you could perform some of the analyses in Forrest et al. paper to test alternative models for the CSC-half-life connection.

We want to thank both reviewers for the insightful comments. Their comments have helped to strengthen as well as clarify our conclusions. For that we are grateful to them.

Originally, we did not comment on the bioRxiv preprint (Forrest et al., 2018), since it was not peer-reviewed and therefore, we were unsure how to discuss the conclusions drawn in a preprint. However, based on the reviewer’s suggestion we have analyzed and compared their data with our data. This analysis is presented in the new Figure 3—figure supplement 1. Based on this analysis in the revised manuscript we have explained in more detail why we conclude that in general amino acids are not playing a central role in mRNA stability in human. However, as we have indicated in the original manuscript, some amino acids might influence mRNA stability (e.g Histidine or Serine, Figure 3 and Figure 6). I have briefly outlined below the major points to ease the review process.

First, we find that the CSC scores of Forrest et al., 2018 preprint highly correlates with our CSC scores, suggesting there are no measurable technical differences between these studies (Figure 3—figure supplement 1A). Second, Forrest et al., 2018 preprint proposed that “amino acid identity are the major determinants of mRNA stability in human”. In fact based on our codon optimality studies in zebrafish and *Xenopus*, we first proposed the idea that amino acids can potentially affect mRNA stability (Bazzini, EMBO, 2016). Therefore, it was natural for us to ask whether a similar relationship between amino acids and mRNA stability also exists in human. Therefore using the parameters we have previously used (Bazzini, EMBO, 2016) and which is also used by Forrest et al., 2018 preprint, we calculated the amino acid stabilization coefficient (ASC), as the Pearson correlation coefficient between mRNA stability and amino acid occurrence (Figure 3—figure supplement 1B). Similar to the Forrest et al., 2018, we also observed that in human cells amino acids composition correlates with mRNA stability and therefore, amino acids can “potentially” be defined as optimal or non-optimal. However, it is crucial to realize that the ASC (amino acids correlation with mRNA stability) is strongly affected by the usage of each synonymous codon. And in both studies find in human (different from zebrafish or *Xenopus*), there are several amino acids (e.g Leucine, Isoleucine, Cysteine, Glutamic acid, Arginine, Asparagine, Phenylalanine, Threonine, Tyrosine, Valine) that are encoded by optimal and non-optimal codons in most of the profiles (Figure 3 and Figure 3—figure supplement 1B). And therefore, the potential amino acid optimality will be dominated by the optimality of the most commonly used codon. For example, for Leucine and Isoleucine, we and Forrest et al., 2018 observed positive ASC scores when all the synonymous codons were used for the calculation, and therefore one could conclude that Leucine and Isoleucine are optimal amino acids (Figure 3—figure supplement 1B). However, as we show in Figure 3—figure supplement 1B when the most abundant codon is not used in the calculation, then the conclusion about the optimality of both amino acids would be the opposite, both amino acids would be defined as Non-optimal amino acids. Therefore, we think that further analysis or conclusions related to intrinsic properties of the amino acids (polar, hydrophobic, etc) cannot be done, because the results will be dominated by the most abundant codon rather than by the amino-acid. The conclusion that amino acids are NOT generally playing a central role in mRNA stability is further supported by two independent observations; first, that synonymous mutations affect mRNAs stability (Figure 3) and second, using our mini-gene reporters we validate that particular amino acids were encoded by optimal and non-optimal codons (Figure 3).

However, there are few amino acids such as Histidine or Serine (Figure 3A), where all the synonymous codons tend to share the same optimality trend (in our and in the Forrest et al., 2018 profiles, Figure 3—figure supplement 1B) and so, the ASC scores are not strongly affected after removing the most used codon (Figure 3—figure supplement 1B). Moreover, for Histidine, we validated that both synonymous codons were non-optimal using mini-genes reporters (Figure 3). We have clarified these issues in the revised manuscript.

Regarding the second issue about the weak correlations between tRNA level and CSC scores, we understand the reviewers’ concerns. While we observed significant correlations between the CSC scores and tRNA level measured by two independent methods, nonetheless the correlations are weak. Moreover, based on the suggestion of reviewer 1, we have assessed the correlation between tRNA level measured by two techniques (Figure 3—figure supplement 1C). The correlation is weak for technical methods indicating that measuring tRNA levels is technically challenging. Following reviewers’ suggestion and based on our new analysis in the revised manuscript we have indicated that relationship between tRNA level and codon optimality is significant but weak and would require further investigation in future.

Your conclusion is expected but has not been shown before and therefore might have turned out differently. Because if this we believe that the paper should be published as "Tools and Resources" paper?

We would prefer to publish it as research article, since we provide insights into mechanism (e.g. poly-(A) tail, level of translation), and the relationship between codon optimality and specific biological process (e.g. virus infection). We have included tables with all the stability profiles (Figure 1—source data 1) and CSC scores (Figure 1—source data 2), so the community can easily use them.

Reviewer #1:

A connection between the codon composition of an mRNA's coding sequence and the half-life of the corresponding transcript has recently been observed in several organisms, including yeast, zebrafish, and *Drosophila*. Similar effects appear to not yet have been published in mammalian cells (a 2018 review (PMID: 29018283) claims that such an effect has been observed in "human, mouse and rat cell lines" but cites only unpublished observation). Using a variety of experimental approaches, Wu et al. demonstrate that codon content and mRNA stability are correlated in several human cell lines.

We have included PMID: 29018283 reference.

The authors globally measure the half-life of mRNAs by performing a time course of RNAseq after shutting off new transcription with actinomycin D treatment. They produce qualitatively similar measurements without incurring potential secondary effects from global transcriptional shutoff using a recently developed metabolic labeling approach. They also measure half-lives of a synthetic library of transcripts in which a large set of endogenous ORFs are inserted between the same 5' and 3' UTRs. Calculating the correlation between the frequency with which each codon occurs in a transcript and that transcript's measured half-life in each experimental approach allows the authors to quantify the effect of that codon on mRNA stability (the codon stability coefficient, or CSC). They observe largely consistent CSC values for each codon across 293T, HeLa, RPE, and K5262 cells and across the three approaches used.To confirm that transcript stability is directly influenced by translation of different codon types (i.e. that this correlation does not simply reflect co-evolution between codon usage and other transcript features influencing stability), the authors follow up by measuring the relatively stability of various reporters in which coding sequences enriched or depleted in "optimal" codons have been placed downstream of fluorescent reporter, separated by a P2A skipping sequence. This allows mRNA abundance to be read out directly by qPCR or indirectly by abundance of the fluorescent protein. Their results support the claim that translation of particular codons can directly stabilize or destabilize transcript. They also demonstrate a positive but weak correlation between CSC and the abundance of each codon's cognate tRNAs from recently published sequencing-based measurements.Major points:The most striking feature of the original observation that codon identity affected transcript stability in yeast was that stabilizing codons consisted almost entirely of codons that had been independently annotated as optimal based on genomic tRNA copy number and codon-anticodon base pairing. […] How strong are Pearson (as opposed to the reported Spearman rank) correlations for 3G and H? (The use of Spearman is reasonable since there is no reason to expect a linear relationship, but the data does not visually suggest a coherent non-linear but monotonic relationship.) How do measured CSC values compare to codon optimality as estimated by relative frequency of usage in the transcriptome or in a subset of highly expressed housekeeping genes?

Please see our comments about the tRNA conclusions above.

For the codon optimality and codon usage, we have added Figure 1—figure supplement 1C. The CSC scores do not present strong correlation with codon usage (Figure 1—figure supplement 1C), from usage frequency of the genome, of the transcriptome (each gene with the weight of mRNA level) and of the top 500 highly expressed genes.

Global measurements of mRNA half-lives by metabolic labeling and sequencing have recently been published in K562s (PMID 29355846) and in mouse embryonic stem cells (PMID 28945705). The manuscript would also be strengthened by both confirming that CSC values inferred from the published K562 data agree with the author's data and by evaluating whether the effect is observed in mouse cells as well.

Based on the reviewer’s comment we have analyzed both datasets and we agree that strengthen our conclusions.

We observed strong correlation between mRNA half-lives calculated with TimeLapse-seq method (Schofield et al., 2018) and our SLAM-seq, both in K562 (Figure 1—figure supplement 1H). Second, we calculated the CSC scores using the mRNA half-lives from TimeLapse-seq (Schofield et al., 2018) and observed strong correlation with our CSC scores (Figure 1—figure supplement 1I).

Finally, we also calculated the CSC scores based on the mouse embryonic stem cells half-lives SLAM-seq profiles (Herzog et al., 2017) and observed strong correlation between the mouse and human CSC scores (Figure 1—figure supplement 1I), indicating a similar codon optimality code in mouse embryonic stem cells.

Reviewer #2:

[…] Major concern:There is an almost identical dataset and analysis that was recently reported, which is not cited or discussed (Forrest et al., 2018). […] It seems to this reviewer that it is important for Wu et al. to discuss the conclusions in Forrest et al., especially when the same data are used to derive different conclusions. Forrest et al. is currently a preprint but it is freely available and the conclusions are highly related – sometimes even involving the same previously published datasets – so it should be discussed.

Please see our comments about the Forrest et al., 2018 preprint above.

Other concerns:1) It is this reviewer's understanding that the ORFeome library used by the authors was generated with invariant 5' and 3' UTRs (Yang et al., 2011). However, the authors indicate that the library has unique barcodes in the 3' UTR, which enables them to perform specific PCR across the barcodes, sidestepping the problem of distinguishing the ORfeome derived reads from endogenous genes during sequencing. This reviewer was unable to find a Material and methods section describing this barcoded library (if the authors have generated one, and a reference if not). The Materials and methods section for sequencing library prep should be elaborated upon.

We ordered the TRC3 ORF Puromycin Arrayed Glycerol Library (Σ-Aldrich), with the barcode for each ORFome mRNA already cloned. We added it in the Materials and method.

2) Furthermore, the authors refer to the ORFeome library as having "common promoter, 5' and 3' UTRs" but if the 3' UTRs have ORF-specific barcodes then they are not common to all library members. How can the authors be sure that including barcodes in the 3' UTR does not affect mRNA stability?

Yes, we cannot exclude the possibility that some barcodes might affect stability (both ways), however there is no relation between barcodes and mRNAs. The barcodes represent less than 1.6% of the nucleotides in the mRNA library. Moreover, we have calculated the decay profile from more than ten thousand mRNAs.

3) It is unclear why the authors attribute the narrow distribution in mRNA levels of the ORFeome to sharing a strong promoter as opposed to having fewer regulatory elements in their untranslated regions – or some combination thereof.

We thank the reviewer for pointing this out. Yes, it is likely a combinatorial effect of promoter, 5’UTR and 3’UTR. We have revised this part in the text to avoid any misunderstanding.

4) The approaches used to generate sequencing libraries in this work (QuantSeq and SLAM-seq and the ORFeome barcoding sequencing) all generate libraries with much lower diversity than standard RNA-seq (since they just sequence the very 3' end of molecules or a barcode region). Because of this, PCR duplicates become more of an issue. How many cycles of PCR were performed? How were PCR duplicates handled?

For ORFome, we have PCR 10, 15 and 20 cycles. With 15 cycles we started seeing a band, then we generated 8 reactions with the exact same condition and pooled them together. For the SLAM-seq, we follow the QuantSeq protocol and do 12 cycle for the PCR.